**communications** engineering

# Reservoir controllers design though robot-reservoir timescale alignment
**Fan Ye** [1], **Arsen Abdulali** [1] ✉, **Kai-Fung Chu** [1], **Xiaoping Zhang**[1,2] **& Fumiya Iida**[1]

Natural behavior emerging in nonlinear dynamical systems enables reservoir computers to control underactuated robots by approximating their inverse dynamics. Unlike other model-free approaches, the reservoir controllers are sample-efficient, meaning a weighted average of the reservoir output can be trained with a limited amount of pre-recorded data. However, developing and testing the reservoir controller relies on repetitive experiments that require researchers' proficiency in both robot and reservoir design. In this paper, we propose a design method for reliable reservoir controllers by synchronizing the timescales of the reservoir dynamics with those observed in the robot. The results demonstrate that our timescale alignment test filters out 99% of ineffective reservoirs. We further applied the selected reservoirs to computational tasks including short-term memory and parity checks, along with control tasks involving robot trajectory tracking. Our findings reveal that a higher computational capability reduces the control failure rate, though it concurrently increases the trajectory-tracking error.

Reservoir computing offers a theoretical framework in which the dynamics of a natural system, the reservoir, can be harnessed for general-purpose computation[1]. The physical characteristics of the reservoir usually cover a certain range of dynamics available to solve a specific task. For instance, the operating ranges of photonics chips[2], memristors[3,4], and oscillators[5] can be used in applications with higher frequencies than those of fluid-based systems[6–9]. However, designing task-specific reservoirs is challenging, as the dynamics emerging within the complex interconnections of the reservoir are not intuitively related to the hyperparameters that researchers can adjust in the system[10].

The importance of reservoir design has been emphasized in the literature for various tasks and applications. The relationship between the existence of a simple deterministic cycle and reservoir memory was discussed in ref. 11. Additionally, researchers in ref. 12 demonstrated that optimal computational performance of the reservoir is achieved at the critical boundary of interconnections. Similarly, the significance of diversifying temporal signals across multiple timescales has been implemented in various ways, for instance, by introducing delayed readout for filter neurons[13], or by stacking multiple layers in a deeper architecture[14,15]. Most of these methods, however, focus more on assessing the effects of various reservoir properties on computational ability. We, on the other hand, believe that the design of the reservoir needs to adequately cover the dynamics or timescales required to approximate the dynamics of a task. In this case, the reservoir computing system can be designed and tested before being deployed to the system. This is particularly important for robotic control, where the reservoir can potentially learn models of robot inverse dynamics from data —a concept we term *reservoir control*.

Reservoir controllers are model-free and computationally efficient robot controllers under offline training[16]. Traditional model-based approaches such as the Linear Quadratic Regulator (LQR)[17–21] and Model Predictive Control[22,23] are predominantly used because of their robustness and predictiveness. The operating range and stability of such systems can be assessed analytically[24]. However, they require formulating mathematical models[25], which is difficult for complex systems. Model-free approaches are compatible with complex systems but require a large amount of data for training. For instance, Feedforward Neural Networks[26–28] and Long-Short Term Memory[29,30] networks-based controllers require a substantial amount of data to update the large set of parameters. One way to train this type of controller is by using reinforcement learning with online training in simulation[31,32]; this method also requires the development of a virtual robot and environment and often necessitates special care to bridge the gap from simulation to reality[33–35]. Reservoir computing, in contrast, uses its inherent nonlinear dynamics to model the inverse dynamics of robots, simplifying this complex process through a straightforward linear regression (LR) of the reservoir output and then realizing the control of different complex system objects.

There were several attempts to apply reservoir computers to varied control tasks, including vehicle navigation[36,37], modulation of chaotic behaviors[38,39], and manipulation of robotic arms[16,40,41]. Several unresolved issues persist in previous studies. First, these studies are often applied for

[1]Department of Engineering, University of Cambridge, Cambridge, UK. [2]School of Electrical and Control Engineering, North China University of Technology, Beijing, China. ✉e-mail: aa2335@cam.ac.uk

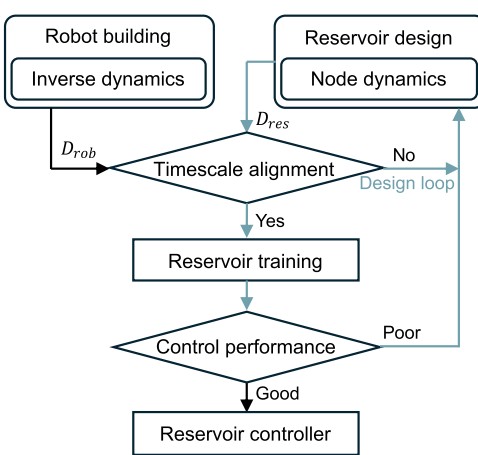

**Fig. 1 | The proposed design method of a reservoir controller.** $D_{rob}$ indicates the robot inverse dynamics, and $D_{res}$ indicates the reservoir dynamics. The timescale alignment test evaluates whether a reservoir's frequency characteristics span beyond the robot's ones, which facilitates filtering out ineffective reservoirs before training and testing on the robot.

fully actuated systems by exerting complete control over the joints and specifying target states without considering higher orders of motion derivatives, such as velocity, acceleration, and jerk[36,37,39–41]. These approaches do not showcase the capability to learn dynamics or highlight any advantages over linear controllers. Second, there is no clear link between the computational ability of the reservoir and the required dynamic range for robotic control, which makes the selection of hyperparameters rely on manual trial and error in[36,37,39–41] or optimization algorithm in[16,38]. This implies testing suboptimal controllers on the physical robots, which makes these approaches less practical for many systems since the cost of each trial is very high and poses the danger of damaging both the robot and the environment.

In this paper, we aim to identify the key criteria for reservoir design in robotic control that ensure the controller can reliably handle the control tasks of nonlinear, unstable, and underactuated robots. The core concept of our design strategy involves synchronizing the timescales of the reservoir with those of the robot, aligning the dynamics bandwidth of each system respectively. Given the rapid training times of the controller, numerous reservoir designs can be generated using random hyperparameters and tested against the proposed criteria. This method allows researchers to quickly identify and eliminate suboptimal controllers, streamline the selection process to just an order of seconds (the algorithm of our design strategy is portrayed in Fig. 1), and select the hyperparameters of the reservoir under the guidance of robot-reservoir timescale alignment. We applied this strategy to the cart-pole system-a nonlinear, unstable, and underactuated robot-and collected the robot timescale and training dataset by moving the cart-pole around the equilibrium point, which can be achieved by manual control of real-world robots or algorithm-based control of simulated cases. The results showed that our approach can eliminate 99% of ineffective reservoirs among all 400 reservoirs with random hyperparameters before they undergo training and testing on the robot, thereby minimizing the risk of causing damage. To ensure stable reservoir behavior across varying initial conditions, we integrate timescale alignment with the echo state property index for effective hyperparameter optimization. In this case, the optimum hyperparameters increased the control success rate of the newly generated reservoirs from 38% to 92% in comparison to timescale alignment solely. The results demonstrated that proposed reservoir controllers can track the cart in complex trajectories and initial conditions unseen during training. This indicates that the reservoir controller is able to extrapolate by emulating the inverse dynamics of the robot. We also conducted an in-depth analysis to assess the effect of the computational properties of the reservoir on benchmark tasks (short-term memory and parity check). We found that a higher computational property reduces the

failure rate of maintaining the pole upright while increasing the trajectory tracking error of the cart. Lastly, the behavior of the reservoir shows that passive nodes (which are actuated only by interconnections, without any direct input signals) contribute more to control tasks than to computational tasks, indicating that robot control tasks rely more on reservoir dynamics than computational tasks. The simulation is written in Matlab (Simulink) and is available at https://github.com/Kyushudy/reservoircontroller.git.

## Results
### Designing a reservoir controller
A reservoir controller uses its reservoir to emulate the inverse dynamics of the robot. This reservoir receives the robot's current state and target state as inputs and outputs the control signal to actuate the robot, see Fig. 2A. There are three steps to building a reservoir controller, including actuation test, reservoir training, and closed-loop control, shown in Fig. 2B.

The control goal of a cart-pole system is to move the cart for a desired trajectory while keeping the pole upwards. This blocks us from randomly exploring the state space by methods like motor babbling since the pole is unstable when positioned upwards. Instead, we picked up an LQR as a benchmark controller to help test the robot dynamics around the balancing point and generate the training data. Given Eqs. (3 and 4), we determine $Q = C'C$, $R = 1$, and $\alpha = 0$ for the benchmark LQR (called LQR1), where $Q$ is the weight matrix for states, $R$ is the weight matrix for inputs, and $\alpha$ is the exponential weight to enhance its aggressiveness. $R$ and $\alpha$ are adjustable to optimize LQRs for further trajectory tracking tasks.

The reservoir controller's performance depends highly on the design of the reservoir. An ideal reservoir emulates the robot's inverse dynamics. The key here is to design a reservoir whose frequency characteristics span beyond that of the robot. In Fig. 3B, we tested the robot's behavior in state space when receiving step and sinusoid references, controlled by LQR1. The step reference requires the cart to stay at $X_r = 0$ m when $t < 15$ s, and move to $X_r = 0.1$ m when $t \geq 15$ s. The sinusoid reference requires the cart to track $X_r$ as a summary of three sine waves with distinct amplitude and frequency. Figure 3C shows the frequency domain graph of the robot under sinusoid references after FFT. All four signals in its state space exhibit three peaks whose frequencies are (0.23Hz, 0.30Hz, 0.56Hz) and maximum amplitudes are (572, 294, 825) with a normalized unit. Similarly, Fig. 3E shows the node behavior of the reservoir under the same input. Figure 3F shows the frequency domain graph of a 10-node echo-state network (ESN) that successfully keeps the pole upwards. All ten signals in its state space exhibit four main peaks whose frequencies are (0Hz, 0.23Hz, 0.30Hz, 0.56Hz) and maximum amplitudes are (1508, 1440, 447, 676) with a normalized unit. Figure 3G shows a 10-node ESN that fails to keep the pole upwards. It has three main peaks whose frequencies are (0.02Hz, 0.23Hz, 0.56Hz) and maximum amplitudes are (5297, 1067, 673).

The timescale alignment test in Fig. 1 helps to predict whether a reservoir controller can keep the pole upwards or not, before testing the controller on the robot. It evaluates whether a reservoir's frequency characteristics span beyond that of the robot, calculated by the containment ratio (CR) in the frequency domain graph. A higher CR indicates the reservoir has a higher potential to emulate the robot's inverse dynamics.

**Test set of reservoirs**. We created 400 reservoirs under random hyperparameters (number of nodes $N = 20$, spectral radius (SR) varying within [0.5, 2], input scaling varying within [0.1, 2], leakage parameter $\tau$ varying within [0.1, 10], regularization coefficient $10^{-8}$, link probability varying within [2/N, 20/N], and bias scaling varying within [0.01, 1]), measured their CR, and tested whether they can keep the pole upwards. Among 400 randomly generated reservoirs, 208 succeed in the control task, while 192 fail. A classifier evaluates whether the reservoir's timescale is aligned to the robot by checking if a reservoir's CR > $CR_T$, the threshold of CR. The null hypothesis is that CR has no effect on control performance, while the alternative hypothesis is that CR is effective in improving control performance. Figure 3H shows the receiver operating characteristic (ROC) curve of this classifier at varying $CR_T$. When

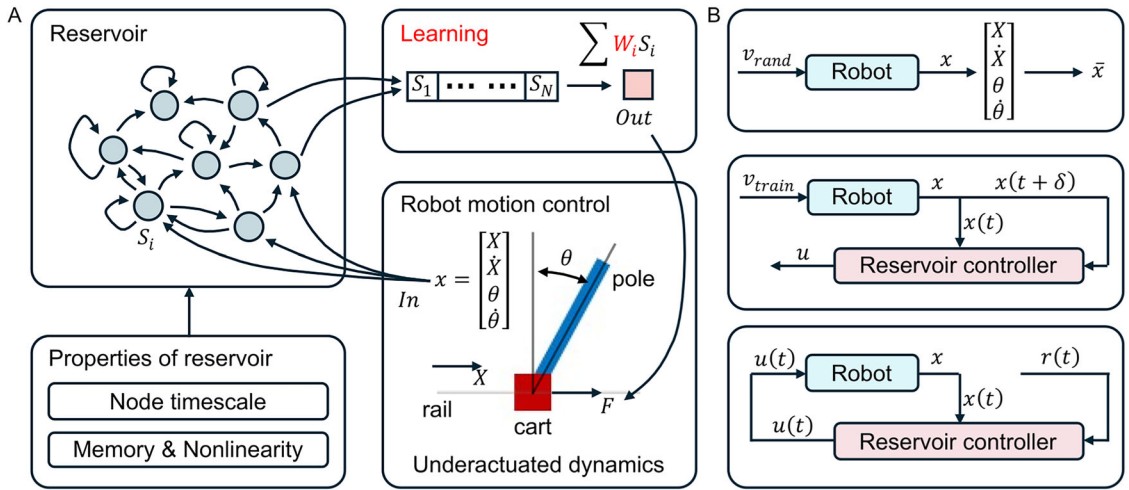

**Fig. 2 | The framework of reservoir control.** A robot controlled by a reservoir computer (**A**) and its three-step training procedure (**B**). The state of the cart pole system is denoted as $\boldsymbol{x}(t) = [X; \dot{X}; \theta; \dot{\theta}]$, while the control reference is $\boldsymbol{r}(t) = [X_r; \dot{X}_r; \theta_r; \dot{\theta}_r]$.

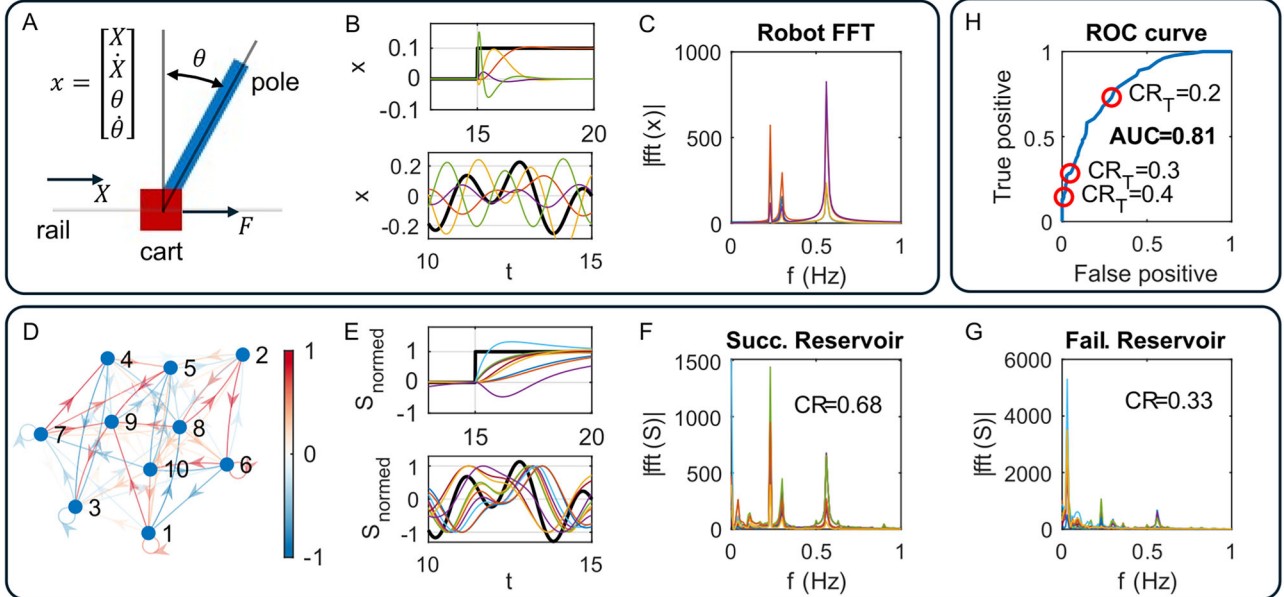

**Fig. 3 | A cart-pole system whose inverse dynamics is potentially emulated by a reservoir.** The cart pole system (**A**), its behavior in state space (colored lines) with step and sinusoid target states (black lines) (**B**), and its behavior under sinusoid target after fast Fourier transform (FFT) (**C**). The 10-node echo-state network (ESN) with interconnections $W_r$ (colored arrows) (**D**), its normalized node behavior (colored lines) under step and sinusoid input (black lines) (**E**). One reservoir succeeded in keeping the pole upwards by covering the robot's frequency characteristics (**F**), while the other one that failed in frequency covering also failed in control (**G**). The receiver operating characteristic (ROC) curve of the timescale alignment test when designing a reservoir (**H**).

$CR_T = 0.2$, the number of (true positive, type I error, type II error, true negative) cases among 400 randomly generated reservoirs is (152, 56, 56, 136), suggesting a true positive rate (TPR = 0.73) and a false negative rate (FPR = 0.29). When $CR_T = 0.4$, the number of (true positive, type I error, type II error, true negative) cases are (30, 2, 178, 190), suggesting (TPR = 0.14) and (FPR = 0.01). The performance of the timescale alignment test is evaluated by the area under the ROC curve (AUC = 0.81). We picked up $CR_T = 0.4$ for the classifier to minimize the false positive rate, since generating a reservoir for the timescale alignment test is easy and fast. Figure 3FG shows that one reservoir, whose CR = 0.68, succeeded in control while another reservoir, whose CR = 0.33, failed in control.

The timescale alignment test also informs the design of our reservoir. To produce reservoirs with a CR > 0.4, we manually adjust the hyperparameters of ESNs to align with the dynamics of the robot, as defined by the following criteria:

- Number of nodes $N = 20$. This value is chosen to be greater than the input dimension (the robot's degrees of freedom) while being sufficiently small to minimize training time.
- Spectral radius SR = 1.1 to satisfy the echo state property[42], ensuring that the reservoir's behavior is not influenced by its initial conditions[43].
- Input scaling 1 for keeping inputs normalized.
- Leakage parameter $\tau = 1$ to ensure the system's frequency characteristics are centered around 1 Hz, facilitating timescale alignment.
- Link probability 10/$N$ to ensure abundant interconnections within the reservoir, preventing overly biased nodes.
- Bias scaling 0.1 for nodes' behavioral diversity.

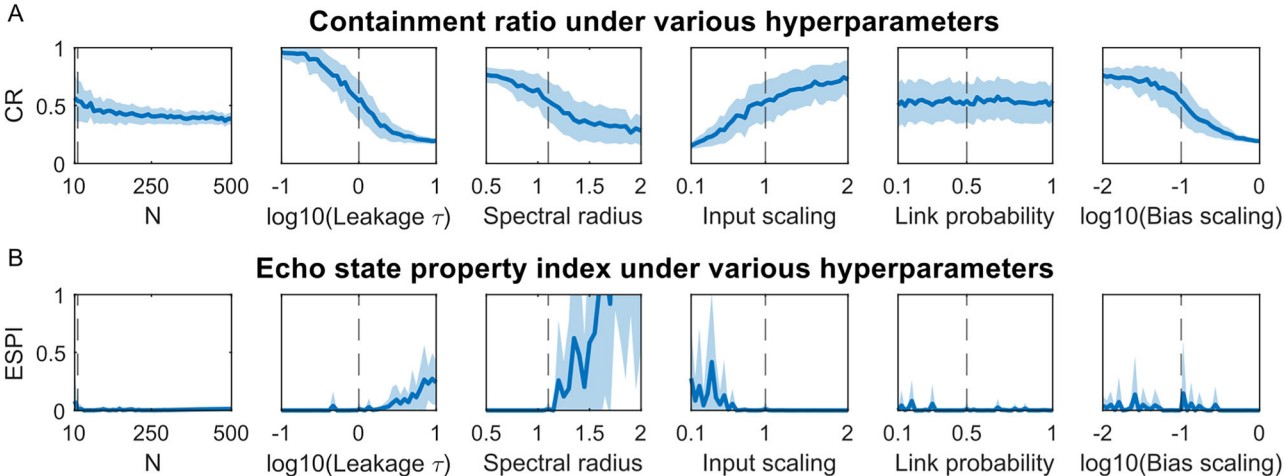

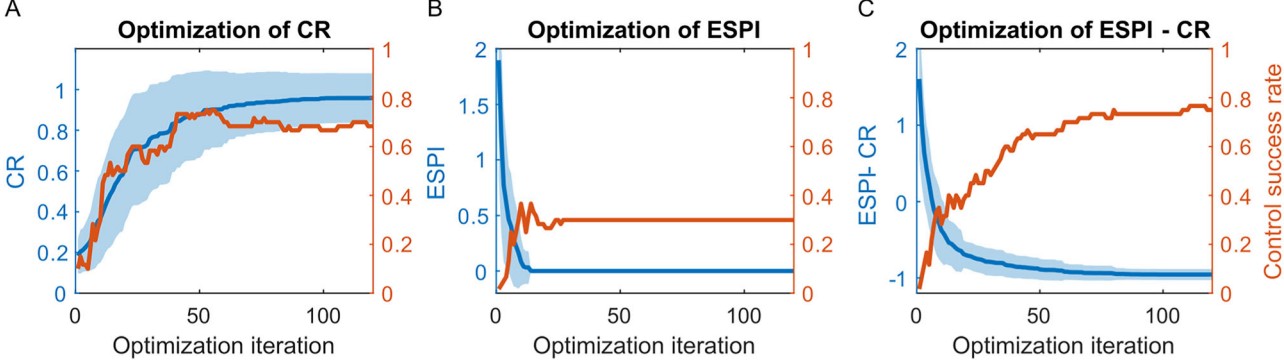

**Fig. 4 | Relationship between reservoir hyperparameters and two indexes describing reservoir dynamics.** The containment ratio (**A**) and the echo state property index (**B**). The default parameter set for our echo-state network (ESN) is ESN20$_s$, shown by dashed vertical lines. In each figure, we vary one hyperparameter and generate 20 ESNs, presenting their mean index values by a dark blue curve and the standard deviation by a light blue shaded area.

**Fig. 5 | Bayesian optimization of reservoir property indexes.** The cost functions are: $-$CR (**A**), ESPI (**B**), and ESPI $-$ CR (**C**), with the goal of minimizing them. The optimization process was conducted for 60 independent trials, all converging by the 120th iteration. Each iteration involves 60 echo-state networks (ESNs) selected based on the lowest cost achieved thus far. The average cost of these ESNs is represented by the dark blue curve, while the associated standard deviation is depicted as a light blue shaded region. Additionally, the red curve illustrates the control success rate of these 60 ESNs at each iteration.

This parameter set is referred to as ESN20$_s$, with a mean CR of 0.55 and a variance of 0.03. To evaluate whether our design strategy holds across different reservoir sizes, we generated additional parameter sets with varying numbers of nodes. These include ESN10$_s$ ($N = 10$, SR = 1.7), ESN50$_s$ ($N = 50$, SR = 1.1), and ESN500$_s$ ($N = 500$, SR = 1.1), whose other hyperparameters are the same as ESN20$_s$.

These parameter sets were manually selected based on general knowledge of the robot's and reservoir's dynamics, with consideration given to the concept of timescale alignment. However, they are not necessarily optimal. Figure 4 investigates the relationship between hyperparameters and CR. Increasing the number of nodes $N$ slightly reduces both the mean and standard deviation of CR, suggesting that a larger reservoir may result in more reliable control. A decrease in $\tau$ significantly increases CR, indicating that our initial choice of $\tau = 1$ was suboptimal, and that a faster-responding reservoir better aligns with the robot's timescale. Increasing $SR$ and bias scaling significantly reduces CR, suggesting that stable ESN behavior is critical for maintaining a high CR. On the other hand, increasing input scaling substantially improves CR, indicating that stronger inputs facilitate better timescale alignment. Lastly, the link probability has no significant effect on CR.

Previous work has also introduced the echo state property index (ESPI)[44] for evaluating ESN dynamics, which, although not specifically designed for robot control, provides valuable comparisons to CR. A lower ESPI suggests reduced sensitivity to initial conditions in the reservoir's

behavior, typically leading to improved task performance. As shown in Fig. 4B, decreasing $\tau$ and SR, or increasing input scaling, reduces the ESPI, indicating the potential for better performance. Parameters such as $N$, link probability, and bias scaling have only minor effects on ESPI. CR and ESPI exhibit similar trends when varying $\tau$, SR, and input scaling, but diverge at bias scaling.

The grid exploration in Fig. 4A indicates the potential existence of parameter sets that enable CR to reach its theoretical maximum of 1. Therefore, hyperparameter optimization for achieving higher CR becomes a benchmark for comparison against our proposed reservoir design method. Previous studies have proposed gradient descent approaches for optimizing reservoir hyperparameters[45,46]. However, in the case of cart-pole control, the random generation of reservoirs based on hyperparameters results in a non-deterministic CR and control success rate. To address this challenge, we employed Bayesian optimization, which accommodates non-deterministic cost functions. The hyperparameter optimization was conducted in 60 individual trials to find the highest CR, with control success rates evaluated accordingly, as shown in Fig. 5A. Each trial comprises 120 iterations to ensure the convergence of CR. The hyperparameter ranges for optimization were defined as follows: number of nodes $N = 20$, spectral radius SR $\in [0.05, 20]$, input scaling $\in [0.01, 20]$, leakage parameter $\tau \in [0.01, 100]$, link probability $\in [2/N, 20/N]$, and bias scaling $\in [0.001, 10]$. At the 1st iteration, hyperparameters were selected randomly, resulting in a mean CR of 0.18 and a

control success rate of 10%. By the 60th iteration, the mean CR increased to 0.91, accompanied by a control success rate of 70%. Beyond the 60th iteration, both metrics showed convergence.

Due to the non-deterministic relationship between hyperparameters and control performance, the optimized parameter set may represent an isolated lucky instance. The optimized ESN achieving the highest CR = 1 was characterized by the following hyperparameters: $N = 20$, SR = 18.99, input scaling = 14.80, leakage parameter $\tau = 0.18$, link probability = 0.81, and bias scaling = 1.44. Although this ESN demonstrated success in the control task during optimization, its large spectral radius (SR > 1) raises concerns about potential instability in control performance when generating additional ESNs with the same hyperparameter set. To investigate this, we generated 100 ESNs using the identified optimal hyperparameters. Among these, only 38% succeeded in the control task, highlighting uncertainties associated with the optimal hyperparameters identified through the optimization process.

These results suggest that the reservoir's echo state property is also needed for consideration during hyperparameter optimization. As shown in Fig. 5B, reducing ESPI from 1.90 to 0.00 increases the control success rate from 2% to 30%. This demonstrates the optimization of ESPI is less effective than optimizing CR in improving control success rates. However, Fig. 5C reveals that a combined approach, using a cost function defined as ESPI–CR to simultaneously minimize *ESPI* and maximize CR, significantly enhances performance. This approach increases the control success rate from 2% to 75%, outperforming optimization based solely on CR. The optimal hyperparameter set identified through this combined optimization process includes: $N = 20$, SR = 6.89, input scaling = 17.17, leakage parameter $\tau = 0.25$, link probability = 0.70, and bias scaling = 0.83. When applied to generate 100 new ESNs, 92% of these reservoirs achieved successful control.

We also provide another manual design strategy of hyperparameters, which ensures both timescale alignment (CR > 0.4) and echo state property, thereby guaranteeing stable reservoir performance in the control task. For example, as described in Section "reservoir design evaluation," the manually designed ESN20$_s$ achieved a 100% control success rate. In terms of generating ESNs that consistently achieve reliable cart-pole control, the manual design method demonstrates a slight advantage over the optimization-based approach.

The central concept of this paper is the introduction of timescale alignment as a guiding principle for reservoir controller design. Therefore, in the following sections, we assess the control performance of manually selected parameter sets (e.g., ESN20$_s$), rather than optimized configurations, to explore how reservoirs can be effectively designed without relying on optimization processes.

## Control performance of reservoir controller

The reservoirs that have passed the timescale alignment test will be applied to control tasks for performance evaluation. The control goal is to enable the cart to track a desired trajectory that includes both position and velocity while holding the pole upwards. Therefore, $r(t)$ can be expressed by $[X_r; \dot{X}_r; 0; 0]$, where $X_r$ is the target position and $\dot{X}_r$ is the target velocity. Figure 6A shows the sine wave tracking task that moves the cart in a sine wave. Figure 6B shows the Lorenz tracking task that moves the cart in a complex trajectory generated by a Lorenz system. The control error is evaluated by dynamic time warping (DTW) to address the tracking performance of $X_r$ and $\dot{X}_r$ while minimizing the effect of time delay. The default robot initial condition is $x(0) = [0\,m; 0\,m/s; 0\,rad; 0\,rad/s]$, and the reservoir initial condition is $s(0) = [0; \cdots; 0]$.

After 100 iterations of reservoir generation under each parameter set (ESN10$_s$, ESN20$_s$, ESN50$_s$, and ESN500$_s$), we picked up the best reservoir controller respectively, called (ESN10, ESN20, ESN50, and ESN500). We also adjusted $R$ and $\alpha$ for optimal LQRs to compare with our proposed reservoir controllers. LQR2 ($R = 9.9967$, $\alpha = 0.4884$) is optimized for minimum DTW error ($e_{dtw}$) in the Lorenz tracking task, and LQR3 ($R = 0.0100$, $\alpha = 1.9993$) is optimized for the sinewave tracking task. Table 1

demonstrates that ESN50 achieves the best performance across both the sine wave and Lorenz tracking tasks among all tested ESNs, and its performance is comparable to the best LQRs in respective tasks.

Figure 6C shows the frequency response of proposed controllers. In this task, the reference trajectory is $X_r = 0.1 \sin(wt)$, where $w$ varies within $[10^{-1}, 10^1]$ rad/s, and $t$ from 0 to 100 s. The error $e_{dtw}$ of trajectory tracking is evaluated at $t \in [10, 100]$ s. As expressed in Section "Training A Reservoir Controller," we train the reservoir with a dataset where the cart-pole system explores its state space at around $w \in [1, 4]$ rad/s. When $w < 1$ rad/s, all the proposed LQR and ESN controllers share similar $e_{dtw}$, ~10 m. When $w > 4$ rad/s, all ESN controllers have higher $e_{dtw}$ than LQR3, but lower than LQR1 and LQR2.

The basin of attraction is another perspective to evaluate the control performance. It is defined as the set of initial conditions for which the controller can keep the pole upwards. As the controller does not directly control the pole, $\theta$, and $\dot{\theta}$ become the prime state for system stability. Figure 6D reveals the basin of attraction of LQR1, ESN10, ESN20, ESN50, and ESN500. The shade of colors reveals the $e_{dtw}$ when the controller performs the sine wave tracking task with different initial states of $\theta$ and $\dot{\theta}$. It shows that when the system states are near the equilibrium point ($\theta \in [-0.2, 0.2]$ rad, $\dot{\theta} \in [-2, 2]\ rad/s$), LQR and ESNs perform well with low $e_{dtw}$, and the control errors raise as the system states reach the boundary of their respective basin of attraction. For the reservoir controllers, an increase in the number of nodes slightly reduces the basin of attraction. LQR1 exhibits the largest basin of attraction, encompassing that of all the ESNs. In addition, Fig. 6D illustrates that ESNs can extrapolate beyond the training data, thereby expanding their basin of attraction.

We generated 154 LQR controllers with varied parameters ($R \in [0.01, 10]$, $\alpha \in [0, 2]$) to compare their task performance, evaluated by $1/e_{dtw}$, against our proposed reservoir controllers. Figure 6E shows that the distribution of LQRs resembles a hyperbolic shape, suggesting that a single LQR is optimized for excelling in only one specific task. In contrast, ESN10, ESN20, and ESN50 perform well across both tasks, while ESN500 exhibits slightly lower performance than smaller ESNs.

## Reservoir design evaluation

We generated 100 random reservoirs under each parameter set (ESN10$_s$, ESN20$_s$, ESN50$_s$, and ESN500$_s$) to evaluate whether good reservoir controllers are replicable based on our design strategy. Figure 7A and Table 2 show the $e_{dtw}$ distribution of generated reservoir controllers in the Lorenz tracking task. An increased number of reservoir nodes will reduce the failure rate but increase control error.

For ESN10$_s$, ESN20$_s$, ESN50$_s$, ESN500$_s$, and LR, the training data volume (as defined by Eq. (9)) and the mean training time of 100 reservoirs are presented in Table. 2 and Fig. 7B. ESN10$_s$ significantly improves computational performance while maintaining a training time comparable to that of LR. Among the ESNs, ESN20$_s$ and ESN50$_s$ strike an economical balance, offering relatively low training times along with strong computational performance.

## Reservoir computational properties

We performed short-term memory (STM) and parity check (PC) tests on ESNs with different numbers of nodes $N$ and spectral radius $SR$ to check their computational properties, shown in Fig. 7DE. The STM task mainly requires the reservoir's memory, and the PC task requires both memory and nonlinearity. Since the reservoir has randomly created interconnections governed by hyperparameters, we created 20 ESNs for each parameter set and computed the mean STM and PC capability. Figure 7DE shows that increasing $N$ from 10 to 500 generally enhances both STM and PC capabilities. Additionally, ESNs with a spectral radius of SR = 1.4 typically outperform those with smaller or larger values of SR.

In Fig. 7C, we plotted all 640 reservoirs that we have created with different hyperparameters no matter whether they pass the timescale alignment test or not, to show the relationship between their computational

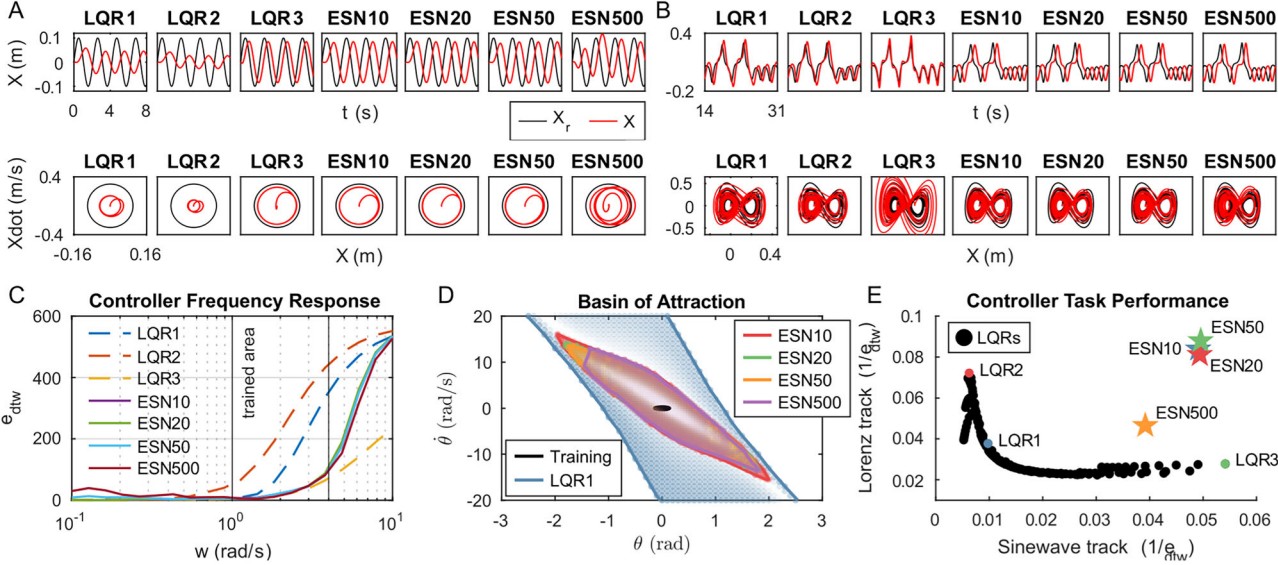

**Fig. 6 | Control performance comparison between linear-quadratic regulators (LQRs) and echo-state networks (ESNs).** The time series and its phase diagram of sinewave track (**A**) and Lorenz track (**B**). The controller's frequency response (**C**), basin of attraction (**D**), and task performance (**E**).

**Table 1 | Control performance data of LQRs and ESNs**

| Controller | Sinewave tracking | Lorenz tracking | Frequency response | | | Basin of attraction | |
|---|---|---|---|---|---|---|---|
| | | | $w = 1.13$ | $w = 3.79$ | $w = 10$(rad/s) | | |
| | $e_{dtw}$ | $e_{dtw}$ | $e_{dtw}$ | $e_{dtw}$ | $e_{dtw}(m)$ | $\theta(rad)$ | $\dot{\theta}(rad/s)$ |
| LQR1 | 102.48 | 26.53 | 9.22 | 334.85 | 534.91 | −6.20 to 6.20 | −32.0 to 33.0 |
| LQR2 | 159.97 | **13.85** | 65.74 | 431.69 | 551.63 | - | - |
| LQR3 | **18.47** | 35.99 | **4.82** | **64.43** | **234.61** | - | - |
| ESN10 | 20.37 | 11.92 | 7.14 | 83.77 | **533.83** | −1.95 to 2.00 | −15.5 to 16.0 |
| ESN20 | 20.25 | 12.32 | 4.97 | 87.19 | 535.80 | −1.80 to 1.80 | −13.5 to 14.0 |
| ESN50 | **20.18** | **11.37** | **4.95** | **81.16** | 534.03 | −1.75 to 1.75 | −13.0 to 13.5 |
| ESN500 | 25.53 | 21.51 | 5.39 | 81.68 | 526.98 | −1.45 to 1.80 | −13.5 to 13.0 |
| Training data | - | - | - | - | - | −0.15 to 0.15 | −0.5 to 0.5 |

The bold values indicate the minimum DTW error or the largest basin of attraction among all LQRs or ESNs, respectively.

properties ($C_{stm}$, $C_{pc}$) and control performance ($e_{dtw}$). We distinguish the reservoirs into two groups, one with high computational property ($C_{stm} + C_{pc} > 7$) and the other with low computational property ($C_{stm} + C_{pc} \leq 7$). Among 91 reservoirs with high computational properties, 1.11% failed to keep the pole upwards, and the mean $e_{dtw}$ among successful reservoirs in the Lorenz control task is 82.00. Among 549 reservoirs with low computational properties, 23.86% failed to keep the pole upwards, and the mean $e_{dtw}$ is 60.85. The reservoir that has the lowest $e_{dtw}$ is at ($C_{stm} = 3.72$, $C_{pc} = 1.81, e_{dtw} = 11.37$ m), and the reservoir that has the best computational performance is at ($C_{stm} = 6.72$, $C_{pc} = 3.21$, $e_{dtw} = 23.58$ m).

**Node behavior of reservoir**

We have shown the definition of a reservoir controller and how it performs in cart-pole control tasks. However, the mechanism of how the reservoir performs these tasks is still unclear. Referring to Eq. (8), the output results from the contributions of bias, input, and states of reservoir nodes. Here, we define each node's contribution to the output as $C_i = \sum_{timesteps} |W_i| \cdot |s_i|$, $i = 1…N$, where $C_i$ is an absolute value that indicates the contribution of node number $i$, summarized in all timesteps. Based on this, $C_{m\sim n}$ indicates the array of $[C_m, …, C_n]$, and $\overline{C_{m\sim n}}$ is the mean of them. The contribution of nodes with inputs indicates a joint

influence of input and reservoir. The performance of ESN50 in different tasks is depicted in Fig. 8.

Figure 8A shows the contribution of nodes when the ESN50 performs different tasks (STM1, STM3, PC1, PC3, sinewave tracking, Lorenz tracking). In the computational property evaluation tasks (STM1, STM3, PC1, PC3), 25 nodes of ESN50 receive inputs, and their average contribution is denoted as $\overline{C_{in}} = \overline{C_{1\sim 25}}$. The remaining 25 nodes are not influenced by external signals, with their average contribution represented as $\overline{C_{pas}} = \overline{C_{26\sim 50}}$. In the cart-pole control tasks (sinewave tracking and Lorenz tracking), 24 nodes of ESN50 receive inputs, contributing a mean value of $\overline{C_{in}} = \overline{C_{1\sim 24}}$, while the other 26 nodes remain unaffected by inputs, contributing $\overline{C_{pas}} = \overline{C_{25\sim 50}}$. The overall ratio of contributions between nodes without inputs and those with inputs ($\overline{C_{pas}}/\overline{C_{in}}$) is illustrated in Fig. 8B. These results suggest that control tasks depend more heavily on the passive dynamics of ESN interconnections compared to STM and PC tasks.

Figure 8C shows the bode diagram of ESN50 when we test the frequency response of its nodes. At the input frequency of 1.08$rad/s$, the magnitude of node spans within $[−25.16, −2.51]dB$, and the phase delay spans within $[−240.72, −12.16]°$. At the input frequency of 9.24 rad/s, the magnitude of node spans within $[−58.66, −21.10]dB$, and the phase delay spans within $[−349.08, −80.20]°$.

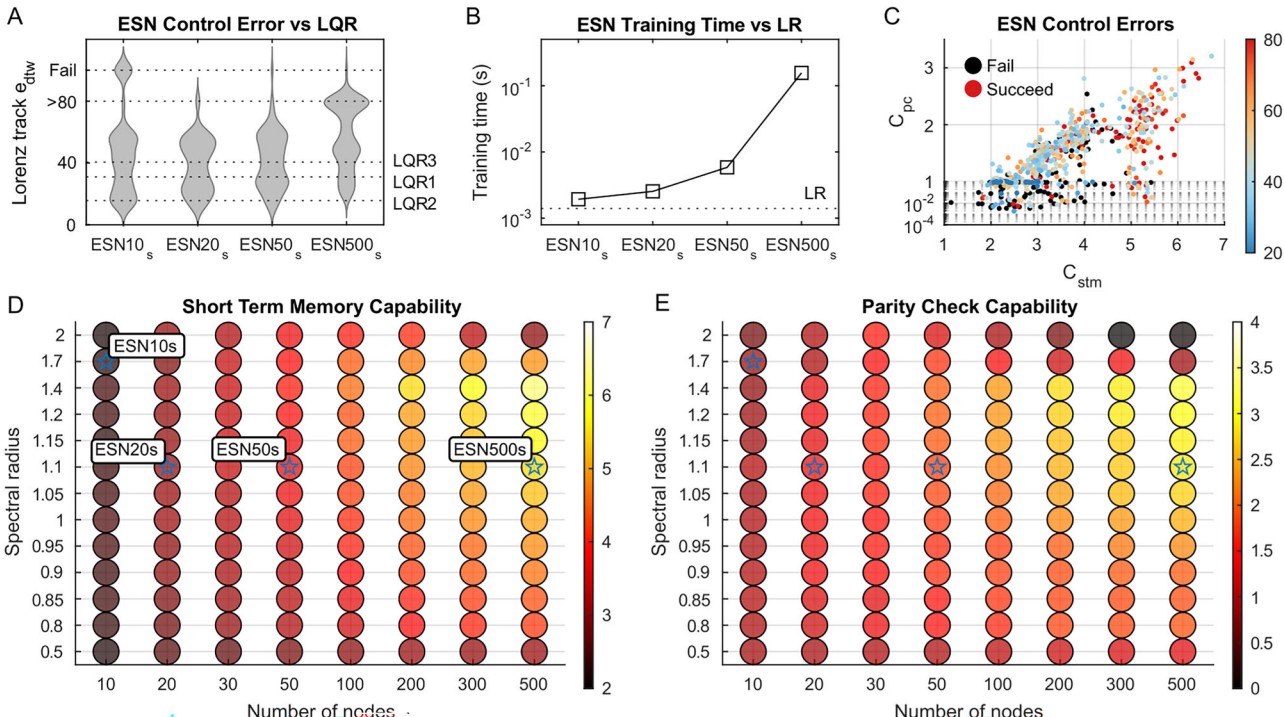

**Fig. 7 | Control performance among echo-state networks (ESNs) with different designed structures, compared with linear-quadratic regulators (LQRs).** The $e_{dtw}$ distribution of Lorenz tracking task among 100 generated reservoirs under different hyperparameter sets (**A**), their mean training time (**B**), the computational property ($C_{stm}$ for short-term memory capability and $C_{pc}$ for parity check capability) of 640 reservoirs and its $e_{dtw}$ shown by colors (**C**), the computational property of reservoirs with different hyperparameters (**D, E**).

**Table 2 | Control and computational performance data of different ESN hyperparameter sets**

| Parameter set | Proportion whose $e_{dtw}$ (m) | | | | Data volume | Training time (s) | $C_{stm}$ | $C_{pc}$ |
|---|---|---|---|---|---|---|---|---|
| | <LQR2 | <LQR3 | >80 | failure | | | | |
| ESN10$_s$ | 11% | 35% | 1% | 14% | 19 × 100,000 | 0.0019 | 2.11 | 0.43 |
| ESN20$_s$ | 9% | 40% | 3% | 0% | 29 × 100,000 | 0.0025 | 3.54 | 1.53 |
| ESN50$_s$ | 5% | 37% | 1% | 1% | 59 × 100,000 | 0.0058 | 4.24 | 2.00 |
| ESN500$_s$ | 0% | 14% | 25% | 1% | 509 × 100,000 | 0.1560 | 6.31 | 2.94 |
| LR | - | - | - | - | 9 × 100,000 | 0.0014 | 0.00 | 0.01 |

## Discussion

In our work, we developed a method of designing a reservoir controller with a highlight of performing a timescale alignment test once a reservoir is generated. The proposed CR is introduced as a quantitative metric to evaluate the suitability of reservoirs for robotic control applications. Experimental results indicate that applying a threshold criterion (CR > 0.4) effectively eliminates 99% of ineffective reservoirs. As solely optimizing CR does not guarantee the reservoir's stable performance, we propose utilizing CR as a supplementary guideline, alongside the echo state property, to guide the selection of reservoir hyperparameters. During the generation of new reservoirs, Bayesian-optimized hyperparameters achieve a control success rate of 92%, whereas human-designed configurations achieve 100%. Our proposed reservoir controller has comparable performance as LQRs in complex trajectory tracking tasks on a cart-pole system.

We used a receiver operating characteristic (ROC) curve to evaluate the performance of the timescale alignment test, which distinguishes reservoirs by CR, measuring whether the frequency characteristics of the reservoir span beyond those of the robot. As shown in Fig. 3H, the area under the ROC curve (AUC) is 0.81, indicating that the classification model of CR performs well in evaluating the suitability of reservoirs for robot control. As the cost of controller trial and danger of damage is very high for robots, $CR_T = 0.4$ (the

threshold of CR) becomes a reasonable choice as it filters out 99% of failure cases while keeping 14% of successful reservoirs (true positive rate TPR = 0.14). In addition, the computational efficiency of offline training, shown in Fig. 7B, that training a small reservoir (number of nodes $N < 50$) requires only $\sim 10^{-3}$ s, makes it easy to generate hundreds of small reservoirs in seconds. Therefore, TPR = 0.14 is acceptable for reservoir controllers.

Due to the uncertain relationship between a reservoir's structure and its hyperparameters, optimizing hyperparameters based solely on CR does not ensure reliable control performance. E.g., the optimal hyperparameters may generate a reservoir with the highest CR and best control performance during optimization, but other reservoirs generated by the same optimal hyperparameters may still have low CR and fail in robot control. This discrepancy arises because such reservoirs may lack the echo state property. As illustrated in Fig. 5A, increasing CR can increase the control success rate from 10% to 70%. However, when the optimal hyperparameters are used to generate new reservoirs, only 38% achieve successful robot control. On the other hand, Fig. 5B demonstrates that optimizing solely for ESPI results in a low control success rate of 30%, as ESPI does not account for the robot-reservoir timescale alignment. To address these limitations, we propose a combined approach that considers both CR and ESPI during hyperparameter design. Figure 5C shows

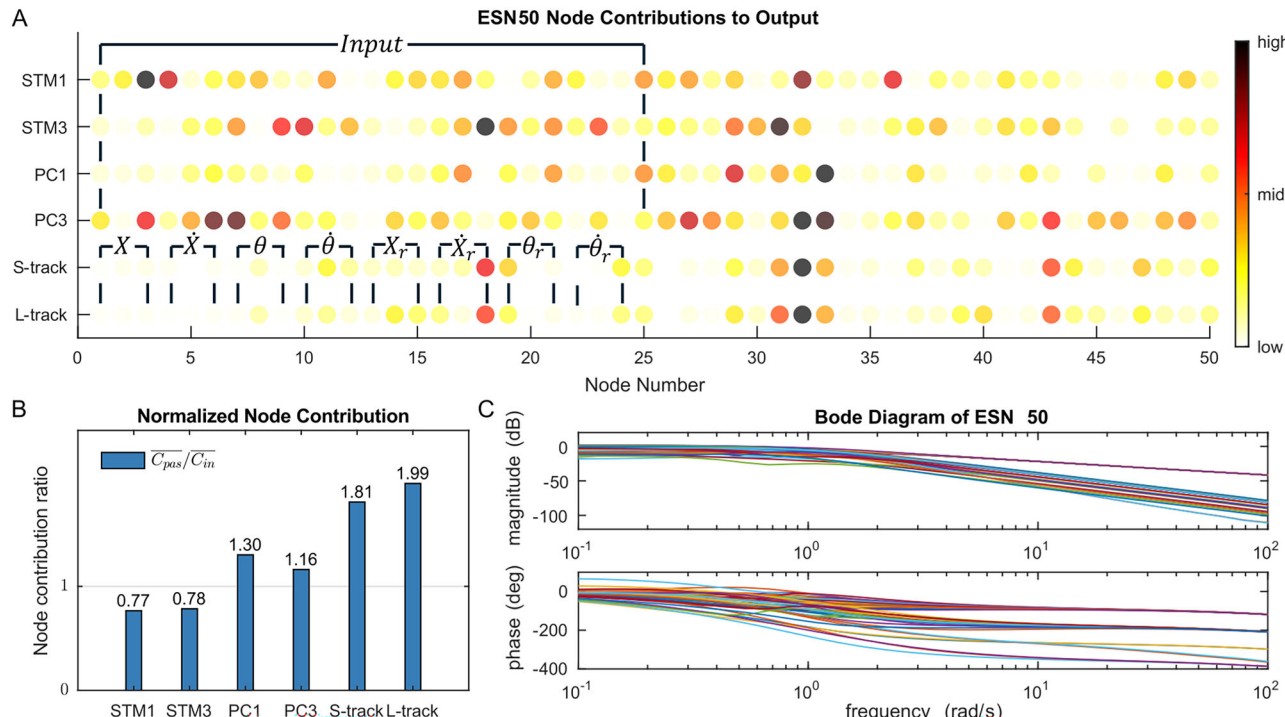

**Fig. 8 | Node behavior of the 50-node echo-state network (ESN50) in different tasks.** The node contribution to output in computational and control tasks (**A**), the comparison of mean contribution between nodes with inputs $\overline{C_{in}}$ and without inputs $\overline{C_{pas}}$ (**B**), the frequency response of ESN50 with 50 colored lines in the bode diagram representing its 50 nodes (**C**).

that by increasing CR and decreasing ESPI through Bayesian optimization, the control success rate increases to 75%, and 92% of the new ESNs generated by the optimum hyperparameter set achieved robot control. Meanwhile, Section "reservoir design evaluation" shows that hyperparameter sets manually designed using the same principles achieve a 100% control success rate.

We then picked up four example reservoirs (ESN10, ESN20, ESN50, ESN500) to evaluate whether reservoir controllers are capable of cart-pole control. Figure 6AB shows that ESNs successfully track the desired complex trajectory in both $X$– $t$ and $\dot{X} - X$ diagram. Figure 6D shows that the reservoir controllers can extrapolate beyond the training dataset, extending the basin of attraction significantly. This indicates that the reservoir successfully emulates the robot's inverse dynamics, shown in Section "Robot: cart-pole system," as achieved through our proposed training methodology. We selected LQRs as benchmark controllers in trajectory tracking tasks to compare with ESNs, and the results in Fig. 6 showed that they have comparable performance. For instance, Fig. 6C shows that the controller frequency response of ESNs is comparable to the best LQR (LQR3) at the frequency area of their training dataset, while at higher frequencies the control errors of ESNs significantly rise to a similar level of ordinary LQR (LQR1). Figure 6E shows that LQRs can only perform well on one of the sinewave and Lorenz track tasks, while ESNs can perform both tasks well. Additionally, a small reservoir ($N < 50$) performs better than a large reservoir ($N = 500$), indicating the bias of each node accumulates as the number of nodes increases, increasing control errors. However, reservoir controllers are more fragile compared to LQRs, shown in Fig. 6D that the basin of attraction of ESNs is smaller than LQR1. The method to increase the robustness of reservoir controllers, e.g,. how to warm up the reservoir before using it for control to reduce the bias caused by initial conditions, is still an open question.

Although example reservoir controllers exhibit attractive performances on robot control, it is unrealistic if we cannot replicate them. We created multiple reservoirs under different parameter sets, e.g., ESN10$_s$, ESN20$_s$, ESN50$_s$, and ESN500$_s$, to test the robustness of our proposed

design method and whether successful reservoir controllers are replicable. Figure 7A shows that the design method works for reservoirs with different numbers of nodes ($N \in [10, 500]$). For ESN10$_s$, ESN20$_s$, and ESN50$_s$, around 10% of generated reservoirs have comparable performance to the best LQR in Lorenz tracking task, indicating that successful reservoir controllers are replicable under our design method. For ESN20$_s$, ESN50$_s$, and ESN500$_s$, the failure rate is around 1%, suggesting that our design method is robust in minimizing the failure rates under different reservoir sizes. However, the failure rate of ESN10$_s$ and the tracking error of ESN500$_s$ are still high, showing the limits of our design method.

An interesting investigation is how the reservoir's computational property contributes to its control performance. The computational properties of ESNs are evaluated by short-term memory ($C_{stm}$) and parity check capability ($C_{pc}$) in Fig. 7DE, and the control performance is evaluated by the DTW error of the Lorenz tracking task ($e_{dtw}$). Generally, the reservoir with more nodes will have higher computational properties. However, Fig. 7A shows no error reduction with the increasing number of nodes. As each reservoir has unique interconnections that result in unique properties, we plot all the generated reservoirs in Fig. 7C to show how their $C_{stm}$ and $C_{pc}$ relate to $e_{dtw}$. In the low computational property area ($C_{stm} + C_{pc} \leq 7$), the failure rate is high (23.86%), while the mean control error is low ($e_{dtw} = 60.85$). On the other hand, in the high computational property area ($C_{stm} + C_{pc} > 7$), the failure rate is low (1.11%), while the mean control error is high ($e_{dtw} = 82.00$). The reservoir with the lowest control error falls in the low computational property area. This result indicates that higher computational property increases the generalization property of the reservoir (i.e., lower failure rate) while reducing the accuracy in trajectory tracking control. The computational complexity of the control tasks may explain this phenomenon. According to Eq. (5), the inverse dynamics of the robot can be expressed using the Euler-Lagrange formulation, which is a second-order nonlinear ordinary differential equation without time delay. Eq. (6) indicates that the control target has a time delay of $\delta = 1$ s. Therefore, a reservoir with a memory span of ~1 s is sufficient for the control task. In comparison,

the timestep for the STM task is 10 s, suggesting that a reservoir with $C_{stm} > 1$ has a memory exceeding 10 s. Since all ESNs in Fig. 7C exhibit $C_{stm} > 1$, having a larger memory may not lead to further improvements in control accuracy.

We explored the node behaviors in computational and control tasks. Figure 8A shows that the node contributions of each node in ESN50 vary from case to case, e.g., STM1, STM3, PC1, PC3, and sinewave tracking control have distinct node contribution distributions. However, both sinewave and Lorenz tracking control exhibit very similar node contributions, indicating that the node contribution remains consistent if the emulated dynamics is similar, regardless of the input variations. Figure 8B shows that the relative contribution of passive nodes $\overline{C_{pas}}/\overline{C_{in}}$ increases as the task shifts from STM and PC tasks to control tasks, indicating that control tasks rely more on the passive dynamics of the reservoir. Figure 8C shows the bode diagram of ESN50, suggesting that the magnitude and phase distribution among 50 nodes help the ESN50's timescale span beyond the robot's. This finding is also found in refs. 13–15, which focuses on learning hidden patterns in complex time series instead of the robot dynamics in this paper.

Although we only tested our method on the cart-pole system, we have developed our approaches to be applicable across different robotic systems. First, the timescale alignment process, as indicated by Eq. (14), estimates the robot's timescale based on the maximum area occupied by its state signals in the FFT graph in Fig. 3C. Regardless of the number of degrees of freedom (DOF) a robot possesses, the same procedure as the cart-pole is applied to get its maximum area, which is used to calculate the CR. Second, the learning task involves an actuation test conducted prior to reservoir training to collect data on the robot's random movements. This data is then used to normalize the robot's state signals, which serve as inputs to the reservoir, see Section "training a reservoir controller". Irrespective of the robot's actual state space, the reservoir operates on a normalized version of the state space, ensuring consistency in its manipulation on different robots. Third, the reservoir hyperparameters can be randomly generated to find reservoirs with a high CR for different robots. The only difference is the minimum number of reservoir nodes. As the input to the reservoir comprises both the robot's current and target states, the reservoir's number of nodes should be at least twice the robot's DOF.

In summary, we developed a design method of reservoir controllers by timescale alignment, which enhances the reservoir's capability to emulate the robot's inverse dynamics. The proposed method can filter out most ineffective reservoir controllers before trying them on the robot, reducing the risk of damaging both robot and environment widely occurring in other model-free control methods. Furthermore, offline training allows us to collect data only once and train hundreds of candidate reservoirs in seconds, making it easy to optimize reservoir controllers. As a supplementary, the achievement of underactuated robot control, the extrapolation of the training dataset, and the contribution of passive nodes show that the reservoir learns the robot's inverse dynamics. We believe this paper benefits the reservoir computing research domain by guiding the design of reservoirs for control purposes.

## Methods
### Robot: cart-pole system
The robot is represented by a cart-pole system. As shown in Fig. 3A, the cart is positioned on a horizontal rail and has one DOF to move on it. The pole is installed on the cart with one DOF to freely rotate around the cart. The cart-pole system is actuated by gravity and an external force that moves the cart on the rail.

The dynamics of the cart and pole can be expressed as:

$$(m + M)\ddot{X} + ml\ddot{\theta}\cos\theta - ml\dot{\theta}^2\sin\theta = F, \tag{1}$$

$$ml\ddot{X}\cos\theta + \frac{4}{3}ml^2\ddot{\theta} - mgl\sin\theta = 0, \tag{2}$$

where $M$ is the mass of the cart, $m$ is the mass evenly distributed on the pole, $l$ is half length of the pole, $X$ is the position of the cart, $\theta$ is the angle of the pole, $F$ is external force, and $g$ is gravity. The value of these parameters is ($g = 9.80665$ m/s$^2$, $m = 0.080$ kg, $M = 0.125$ kg, $l = 0.1$m). By linearizing Eqs. (1 and 2) around the fixed point ($X = 0, \dot{X} = 0, \theta = 0, \dot{\theta} = 0$), we get the state-space model below:

$$\begin{bmatrix} \dot{X} \\ \ddot{X} \\ \dot{\theta} \\ \ddot{\theta} \end{bmatrix} = \underbrace{\begin{bmatrix} 0 & 1 & 0 & 0 \\ 0 & 0 & \frac{-3mg}{m+4M} & 0 \\ 0 & 0 & 0 & 1 \\ 0 & 0 & \frac{3(m+M)g}{(m+4M)l} & 0 \end{bmatrix}}_{A} \begin{bmatrix} X \\ \dot{X} \\ \theta \\ \dot{\theta} \end{bmatrix} + \underbrace{\begin{bmatrix} 0 \\ \frac{4}{m+4M} \\ 0 \\ \frac{-3}{(m+4M)l} \end{bmatrix}}_{B} u, \tag{3}$$

$$y = \underbrace{\begin{bmatrix} 1 & 0 & 0 & 0 \\ 0 & 0 & 1 & 0 \end{bmatrix}}_{C} \begin{bmatrix} X \\ \dot{X} \\ \theta \\ \dot{\theta} \end{bmatrix} + \underbrace{\begin{bmatrix} 0 \\ 0 \end{bmatrix}}_{D} u, \tag{4}$$

where $\mathbf{x}(t) = [X; \dot{X}; \theta; \dot{\theta}]$ is the state space of the cart-pole, shown in Fig. 3A.

As described above, the reservoir computer aims to control the cart-pole system by emulating its inverse dynamics. Based on Eqs. (1 and 2), the cart-pole dynamics can be represented using the Euler-Lagrange formulation as follows:

$$\begin{bmatrix} F \\ 0 \end{bmatrix} = \begin{bmatrix} m+M & ml\cos\theta \\ ml\cos\theta & \frac{4}{3}ml^2 \end{bmatrix} \underbrace{\begin{bmatrix} \ddot{X} \\ \ddot{\theta} \end{bmatrix}}_{\ddot{q}} + \begin{bmatrix} 0 & 0 \\ 0 & -ml\dot{\theta}\sin\theta \end{bmatrix} \begin{bmatrix} \dot{X} \\ \dot{\theta} \end{bmatrix} + \begin{bmatrix} 0 \\ -mgl\sin\theta \end{bmatrix} \tag{5}$$

where $\ddot{q}$ is the second derivative of system state $q$ by time ($q = [X; \theta]$). In Eq. (5), only $F$ and $\ddot{q}$ are unknown. Inverse dynamics controllers compute $F$ by providing a mathematical expression for $\ddot{q}$. Suppose we have a desired trajectory $p$ (where $p = [X_r; \theta_r]$), $\ddot{q}$ can be determined based on the desired system dynamics. Below are some e.g.,[47,48]:

$$\begin{cases} \ddot{q} - \ddot{p} = 0 & \text{Acceleration Tracking}, \\ \ddot{q} - \ddot{p} + K_d(\dot{q} - \dot{p}) + K_p(q - p) = 0 & \text{Computed Torque Control}, \\ q(t+\delta) = p(t), \ \dot{q}(t+\delta) = \dot{p}(t) & \text{Reservoir Control}, \end{cases} \tag{6}$$

where the acceleration tracking method minimizes the error between $\ddot{q}$ and $\ddot{p}$, the Computed Torque Controller additionally considers the derivative and proportional part by $K_d$ and $K_p$, and the reservoir controller tries to move the robot from current state $q(t)$ to target state $p(t)$ within a short time interval $\delta$. All these control approaches can be generalized by the function $F = G(q, p)$, a general expression of inverse dynamics.

### Reservoir computer: echo-state network (ESN)
ESN is a type of reservoir computer whose reservoir is an N-node pre-defined RNN, shown in Fig. 3D. Assuming an ESN has $N_{in}$ number of input signals and $N_{out}$ number of output signals, we can describe the ESN by the differential equation:

$$\tau\dot{\mathbf{s}} = -\mathbf{s} + \tanh(\mathbf{W_{in}In} + \mathbf{W_r s} + \mathbf{b}), \tag{7}$$

$$\mathbf{Out} = \mathbf{W_{out}}[1; \mathbf{In}; \mathbf{s}], \tag{8}$$

where $\tau$ is the leaking rate which determines the behavioral timescale of the reservoir, $s$ (dimension $N \times 1$) is the state of nodes in RNN, $tanh$ is a hyperbolic tangent function, $W_{in}$ ($N \times N_{in}$) is the input weight matrix, $W_r$ ($N \times N$) is the recurrent weight matrix, $W_{out}$ ($N_{out} \times 1 + N_{in} + N$) is the output weight matrix, and $b$ ($N \times 1$) is the bias for each node. As a timeseries with $N_t$ timesteps, $In$ ($N_{in} \times N_t$) is the input signal, and $Out$ ($N_{Out} \times N_t$) is the output signal at each timestep. In Eq. (8), $[1; In; s]$ represents the combined contribution of bias, input, and reservoir states to generate the output. This equation regulates $s$ between $[-1, 1]$ and guarantees the nonlinear behavior of nodes. Note that the values in $W_{in}$, $W_r$, $b$ are fixed, while only $W_{out}$ is adjusted by ridge regression for desired outputs when training a reservoir computer.

According to ridge regression, $W_{out}$ in Eq. (8) can be given by:

$$W_{out} = T_{Out}[1; In; s]^{\top}([1; In; s][1; In; s]^{\top} + \beta I)^{-1}, \quad (9)$$

where $\beta$ is the regularization coefficient, $T_{Out}$ is the desired output, and $I$ is the identity matrix.

Comparing against a standard LR which finds the linear combination of $[1; In]$, an ESN uses the data of $[1; In; s]$ to approximate $Out$ in Eq. (8). This indicates that ESN leverages the reservoir to better emulate a dynamical system, despite it has a similar training method with LR. In addition, this grants ESN with a similar computation time as LR. The comparison between LR and ESN is presented in Section "Results".

Two capability tasks are developed in previous research[7,12] to evaluate the reservoir performance of an ESN. The first task is called the short-term memory (STM) task, which requires the output of ESN to be a delayed version of the binary input. Therefore, the STM task checks the memory capability of the reservoir, and the longer delay indicates a better memory capability. The second task is called the parity check (PC) task, which requires the ESN to tell if the previous binary inputs have an odd number of "1". The PC task checks both the memory and nonlinearity capability of the reservoir.

The STM task can be expressed as:

$$T_k = In_{k-\tau_B}, \quad (10)$$

where $T_k$ is the target at timestep $k$, and $\tau_B$ is the delay. STM1 indicates an STM task with a delay of ($\tau_B = 1$), STM3 ($\tau_B = 3$), and so on. A larger $\tau_B$ requires better memory capability.

The PC task can be expressed as:

$$T_k = Q\left(\sum_{i=0}^{\tau_B} In_{k-i}\right), \quad (11)$$

$$Q(x) = \begin{cases} 0 \ (x \equiv 0 \bmod 2) \\ 1 \ (\text{otherwise}). \end{cases} \quad (12)$$

Similarly, PC1 indicates a PC task with a delay of ($\tau_B = 1$), and a larger $\tau_B$ requires better memory and nonlinearity capability. For each run, the ESN is simulated for 2000 timesteps with $dt = 10$ s. The first 200 timesteps are discarded to reduce the effect of the reservoir's initial condition, and the next 1800 timesteps are used for training or performance evaluation. The reservoir is first trained by a randomly created binary input (2000 timesteps) and evaluated on another randomly created binary input (2000 timesteps).

In Section "Result," we performed STM and PC tasks on ESNs with different hyperparameters $N$, the number of nodes in the reservoir, and $SR$, the spectral radius used to randomly create $W_r$. The reservoir performance in STM and PC tasks are evaluated by $\tau_B$-delay capability $C(\tau_B)$, expressed as:

$$C(\tau_B) = \frac{cov^2(Out, T)}{\sigma^2(Out)\sigma^2(T)}. \quad (13)$$

$C(\tau_B)$ varies between $[0, 1]$ and equals 1 if $Out = T$. As most ESNs perform well in STM1 and PC1 tasks but poor in STM10 and PC10 tasks, the STM capability $C_{stm}$ of a reservoir is a sum of $[C_{stm}(1), C_{stm}(2), \ldots C_{stm}(10)]$. Similarly, $C_{pc}$ represents the PC capability of the reservoir. Both $C_{stm}$ and $C_{pc}$ vary between $[0, 10]$ and higher numbers show better performances in STM and PC tasks.

## Timescale alignment test: evaluation of reservoir dynamics

The reservoir is evaluated by a timescale alignment test once generated. It checks whether a reservoir's frequency characteristics span beyond the robot's, calculated by a CR in the frequency domain graph. We test the frequency characteristics of the cart-pole system by exploring its state space while keeping the pole upwards by LQR1. As shown in Fig. 3B, the reference of LQR1 is the position of the cart ($r(t) = [X_r; 0; 0; 0]$), where $X_r$ is a combination of three sinewave expressed by Eq. (17), with parameters of ($A_1 = -0.049$, $A_2 = -0.090$, $A_3 = 0.151$ m, $w_1 = 1.863$, $w_2 = 1.444$, $w_3 = 3.540$ rad/s). The simulation runs for 100 s to collect the robot behavior in state space ($x = [X; \dot{X}; \theta; \dot{\theta}]$), and the collected signals are processed by Fast Fourier Transform (FFT) to get the frequency characteristics of each signal ($fft(x) = [fft(X); fft(\dot{X}); fft(\theta); fft(\dot{\theta})]$), shown in Fig. 3C. Then, the robot's frequency characteristics is the maximum amplitude of all the signals:

$$|fft(rob)| = \max(|fft(X)|, |fft(\dot{X})|, |fft(\theta)|, |fft(\dot{\theta})|). \quad (14)$$

Similarly, the reservoir receives the same input as the robot ($In = X_r$), and output the reservoir states ($s = [s_1, \ldots, s_N]$, where $N$ is the number of nodes), see Fig. 3E. After FFT, we get the reservoir's frequency characteristics as:

$$|fft(res)| = \max(|fft(s_1)|, \ldots, |fft(s_N)|). \quad (15)$$

The CR is defined to evaluate the similarity of $|fft(rob)|$ and $|fft(res)|$. However, as the magnitudes of robot and reservoir states are not comparable, both of them are normalized by the maximum amplitude ($\overline{|fft(rob)|} = |fft(rob)|/\max(|fft(rob)|)$, and $\overline{|fft(res)|} = |fft(res)|/\max(|fft(res)|)$). Then, CR is computed by:

$$CR = \frac{\sum_{f=0}^{f=1} \min(\overline{|fft(rob)|}, \overline{|fft(res)|})}{\sum_{f=0}^{f=1} \overline{|fft(rob)|}}, \quad (16)$$

where $f$ is the frequency varying from 0 to 1 Hz. When $CR < CR_T$, the timescale alignment test rejects the reservoir; otherwise, the reservoir is accepted and proceeds to the control task.

*Echo state property index* (*ESPI*): The *ESPI* evaluates the sensitivity of a reservoir's behavior to initial conditions under identical input sequences. This concept aligns with the conditional Lyapunov exponent (CLE)[49,50]. For simplicity and clarity, we adopted the measure called *ESPI*. Similar to the computation of *CR*, the reservoir is provided with input data ($In = X_r$) and produces node states ($s = [s_1, \ldots, s_N]$) for 100 seconds. The baseline behavior, denoted as $s^0$, starts from a default initial state $s(0) = [0, \cdots, 0]$, while 10 alternative behaviors, $s^1 \cdots s^{10}$, are initiated from randomly selected initial states uniformly distributed between $[-1, 1]$. The difference between each alternative behavior and the baseline is quantified using mean-squared error ($e_{behavior}^i = MSE(s^i, s^0)$, where $i = 1, \cdots, 10$), with a washout period of 50 s. In this case, the behavior error reflects the lasting influence of initial conditions longer than 50 s. The *ESPI* is calculated as the average of the behavior errors ($ESPI = \overline{[e_{behavior}^1, \cdots, e_{behavior}^{10}]}$). A low *ESPI* indicates that the reservoir exhibits stable behavior under the same input, regardless of the initial state, thus supporting reliable computation. This index was introduced by ref. 44.

**Article**

## Training A reservoir controller

Three steps of training a reservoir controller need to be taken to achieve model-free control on a robot with unknown dynamics, shown in Fig. 2B:

1. Actuation test. The robot is actuated by a random input signal $v_{rand}$ and outputs its states. The reservoir controller collects the magnitudes of the states $\boldsymbol{x}$ and normalizes each element of the state matrix by the absolute median.
2. Reservoir training. The robot is actuated by an input signal $v_{train}(t)$ aiming to explore its state space. Then its current states $\boldsymbol{x}(t)$ and future states $\boldsymbol{x}(t + \delta)$ are collected for reservoir training. The reservoir computer tries to map current states $\boldsymbol{x}(t)$ and future states $\boldsymbol{x}(t + \delta)$ to current actuation forces $v_{train}(t)$ by LR.
3. Closed-loop control. The reservoir input of future states $\boldsymbol{x}(t + \delta)$ is replaced with desired states of the plant $\boldsymbol{r}(t)$, and the reservoir output $u(t)$ serves as a control signal for the robot.

The reservoir controller controls the robot in a closed loop by receiving current states $\boldsymbol{x}(t)$, target states $\boldsymbol{r}(t)$, and generating control signal $u(t)$. The states of the robot $\boldsymbol{x}(t)$, $\boldsymbol{r}(t)$ can include information about position and velocity, which helps the robot track a complex trajectory.

Here are the details about how we train ESNs for cart-pole control. In the actuation test, the cart-pole system is actuated by a random signal for 100 s to collect the absolute median of states $[|\bar{X}|; |\dot{X}|; |\bar{\theta}|; |\dot{\theta}|]$ to normalize the input into the reservoir. In the reservoir training, the data of LQR1 tracking a sinewave for 100 s with $dt = 0.01s$ are collected to get the combination of $\boldsymbol{x}(t)$, $\boldsymbol{x}(t + \delta)$, and actuation forces $v_{train}(t)$, where $\delta = 1s$. The sinewave-based reference is expressed by:

$$X_r = A_1 \sin(w_1 t) + A_2 \sin(w_2 t) + A_3 \sin(w_3 t), \qquad (17)$$

where $(A_1, A_2, A_3, w_1, w_2, w_3)$ are random values varying within ($[-0.4, 0.4]$, $[-0.2, 0.2]$, $[-0.3, 0.3]$, $[1.8, 2.2]$, $[1.26, 1.54]$, $[3.06, 3.74]$), whose units are $m$ for $A$ and $rad/s$ for $w$. The setting of $X_r$ keeps $X$ random and dynamic for data collection, and this process runs 10 times to build the dataset for (100/ 0.01*10 = 100,000) timesteps. The dataset is only collected once and used to train all the ESNs. Then, ESN receives $\boldsymbol{x}(t)$ and $\boldsymbol{x}(t + \delta)$, and reduces the error between $u(t)$ and $v_{train}(t)$ by ridge regression.

In closed-loop control, the reservoir controller is tested on two control tasks with different target states $\boldsymbol{r}(t)$. The first control task is sinewave tracking whose $\boldsymbol{r}(t) = [X_r; 0; 0; 0]$, where $X_r = 0.1 \sin(3t) \, m$ at $t \in [0, 40]s$. The second control task is Lorenz tracking whose $\boldsymbol{r}(t) = [X_r; \dot{X}_r; 0; 0]$. $X_r$ and $\dot{X}_r$ are collected from a Lorenz system with parameters ($\rho = 28$, $\sigma = 10$, $\beta = 8/3$). This guarantees that $X_r$ and $\dot{X}_r$ represent a reasonable complex trajectory and are achievable by a dynamical system. The trajectory of $X_r$ and its phase diagram ($\dot{X}_r - X_r$ plot) are shown in Fig. 6AB.

The error of trajectory tracking is evaluated by dynamic time warping ($e_{dtw}$). Given the desired trajectory $X_r(t)$ and the actual robot trajectory $X(t)$, we have the distance matric:

$$\boldsymbol{d_{m,n}} = X_r(m) - X(n), \qquad (18)$$

where $X_r(m)$ indicates the $m$th sample of $X_r$, and $X(n)$ indicates the $n$th sample of $X$. Then, the trajectory error is calculated by:

$$e_{dtw} = \min\left(\sum_{m \in ix, n \in iy} \boldsymbol{d_{m,n}}\right), \qquad (19)$$

where $ix$, $iy$ are two monotonically increasing sequences of the same length. In this way, we calculate the trajectory error of $X_r(t)$ and $X(t)$ while ignoring the possible time delay between them. Ideally, $e_{dtw} \simeq 0$ if $X_r(t) = X(t + \delta)$, which means the robot position precisely follows the reference with a time delay $\delta$.

## Data availability

MATLAB simulation data and a supplementary movie that support the findings of this study have been deposited in https://github.com/Kyushudy/reservoircontroller.git.

## Code availability

All the codes are written in MATLAB (Simulink), available at https://github.com/Kyushudy/reservoircontroller.git.

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

## Acknowledgements
This work was supported by a Huawei HiSilicon Scholarship. This project has received funding from the European Union's Horizon 2020 research and innovation programme under the Marie Skłodowska-Curie grant agreement No 101034337.

## Author contributions
F.Y., A.A., and F.I. developed ideas and designed methodology. F.Y. wrote codes, conducted the experiments, collected data, and wrote the first draft. A.A., K.F.C., X.P.Z., and F.I. proofread the manuscript and contributed to discussions. All authors made substantial contributions to the paper's content and have read and approved its publication.

## Competing interests
The authors declare no competing interests.
