## [Transparent Peer Review File · Communications Engineering]

Reservoir Controllers Design Though Robot-Reservoir Timescale Alignment

Corresponding Author: Dr Arsen Abdulali

Version 0:

Reviewer comments:

Reviewer #1

(Remarks to the Author)

This study proposed a robot controller design using reservoir computing where reservoir parameters are screened so that the frequency of the robot (task) movements matches that of the reservoir dynamics, which is termed as robot-reservoir timescale alignment. From the data obtained when the robot was controlled by LQR, the reservoir learned the inverse dynamics of the robot, resulting in a controller that is more robust than LQR. Although a method for constructing a controller using a reservoir has already been proposed [16, Zhai et al., 2023], the idea of aligning the timescale of the controller (reservoir) and the controlled object (robot) is interesting. However, the timescale of the reservoir is mainly determined by the leakage parameter or time constant of unit dynamics, and the relationship between it and the alignment is unclear. A comparison with a method that optimize it directly is also needed. The following are major comments.

- (1) The definition of inverse dynamics is unclear. Please clarify the learning task of the reservoir so that it can be understood by non-specialists in robotics.
- (2) In this study, timescale alignment was proposed to simplify the adjustment of reservoir parameters. However, a method of optimizing the reservoir parameter (leakage parameter and spectral radius) using a stochastic gradient descent method has also been proposed [A]. Identify the advantages of the proposed method compared to this existing method.
- (3) As mentioned above, the timescale of the reservoir strongly depends on the leakage parameter. The relationships between reservoir parameters and the containment ratio (CR) should be analyzed.
- (4) The spectral radius is the most important parameter for the performance of the reservoir. For general echo state networks, the best parameter value is about 1.0. However, as far as Fig. 5DE is concerned, its parameter search tick range is 0.3, which is too coarse. The spectral radius dependence of its performance is particularly strong as the network size increases. A more detailed parameter search, including around 1.0, is needed.

[A] Jaeger et al., "Optimization and applications of echo state networks with leaky-integrator neurons," *Neural Networks*, 20(3), 335-352, 2007.

Reviewer #2

(Remarks to the Author)

This paper investigates a reservoir computing approach for robot control, proposing a novel scheme called a timescale alignment. This scheme basically compares the FFT spectrum of the timeseries obtained by the target robotic systems and that obtained by the reservoir states, which takes the robotic timeseries as input, using the heuristic measure called containment ratio (CR). As a result, the scheme suggests that reservoirs that align their timescale with the robot show good control performance. Further systematic analyses of timescale aligned ESN are conducted, including measuring its STM and PC capacities. Overall, the topic is timely and the study includes interesting contribution to the field. Thus, I recommend the paper be accepted after addressing the following points (including major and minor points).

1. It is unclear how the hyperparameters of ESN are updated through the timescale alignment procedures. It is good to use CR to compare, but are these procedures performed through grid search of hyperparameters or are there any update rule proposed, such as gradient descent type scheme? I couldn't find clear descriptions about this point (or, if it was described somewhere, please stress it).
2. One important property in reservoir computing is the synchronization of input stream with reservoir dynamics, which is termed generalized synchronization [2a] or echo state property (ESP) [2b]. This property guarantees the reproducible response of the reservoir against identical input stream and is essential for reliable computation. It would be beneficial for readers if the authors investigate the relationship between timescale alignments and ESP. ESP can be characterized by

ESP index using random input usually [2c] (In the current paper, we can use the input stream, coming from the robotic system, instead). For example, there should be some parameter regions that has ESP but less timescale alignment (opposite can also occur). This analysis will provide fundamental and interesting aspects to the characteristic of reservoir dynamics. Furthermore, it would be better to analyze conditional largest Lyapunov exponent of ESNs, if possible, which provides more accurate condition of the reservoir dynamics.

[2a] Lu, Z., Hunt, B. R., & Ott, E. (2018). Attractor reconstruction by machine learning. *Chaos: An Interdisciplinary Journal of Nonlinear Science*, 28(6).

[2b] Yildiz, I. B., Jaeger, H., & Kiebel, S. J. (2012). Re-visiting the echo state property. *Neural networks*, 35, 1-9.

[2c] Gallicchio, C. (2018). Chasing the echo state property. arXiv preprint arXiv:1811.10892.

3. It would be helpful for readers if the authors attached the supplementary video on how the reservoir works on robot control.

4. The term "edge of chaos" is not used so much in recent days in reservoir computing community because it is misleading in multiple aspects. Please check Ref. [3a] for details.

[3a] H. Jaeger (2021): Foreword to the book *Reservoir Computing: Theory, Physical Implementations, and Applications* (K. Nakajima and I. Fischer, eds.), Springer Nature Singapore

5. Figure references are sometimes incorrect. For example, in Section 4.2, first sentence, Fig. 3C should be Fig. 2A? (or, at least, it should not be Fig. 3C). In Section 4.3, sixth sentence, is Fig. 3E correct? Please again check the figure reference in detail.

Reviewer #3

(Remarks to the Author)

In this submission the authors present an approach for utilising echo-state network (ESN) reservoirs for model-free control of a robotic system. By implementing a validation algorithm that tests the randomly created reservoirs for suitable matching of reservoir to the timescale of the robot motion the authors are able to focus on better performing reservoirs. The authors show that the ESN reservoir is able to control the agent within the cart-pole task, where the cart is required to follow a desired trajectory while maintaining the pole vertical.

The major claims of the paper are that timescale alignment of the reservoir and model dynamics can help filter out unsuitable reservoirs and that ESN reservoirs can be used as a robotic controller comparable to conventional models such as linear quadratic regulators (LQR). This is only demonstrated on the cart-pole task and performance appears to degrade with increasing reservoir size. Overall, whilst only a single demonstration of the reservoir controller, this work does present a novel filtering strategy and evaluation of the reservoir performance. Some aspects of the work needs further clarification as follows:

- The demonstration of the methodology is only on a task with a single joint and it is unclear how it would translate to more complicated tasks, e.g if the cart pole had multiple joints. Have the authors considered other robotic tasks and could they discuss this translation to more complication systems?

- Could the authors clarify the computational complexity of the task? The small 10 node ESNs work well for the task, better than the larger networks, and usually with lower STM or PC metrics and it would useful to have a deeper discussion on this. If the network simply needs to recall a step or two of input with minimal non-linearity to provide the correct signal to the cart-pole then it means it is harder to generalise the conclusions to broader control tasks. The concept that larger reservoirs inhibit the model-free control due to a larger transformation to invert relates to this (for example as discussed in ref 39) but further discussion and analysis of this would be beneficial.

- The timescale alignment evaluates the containment ratio of the reservoirs timescales to span that of the robots. This doesn't appear to consider the placement of the peaks in the reservoir frequency response. If the reservoirs fundamental frequency is close to be off resonance of the robot is there poor matching of the dynamics? Would it be better to test reservoirs to have a broader frequency response? While the method should be of interest to other researchers it needs some deeper analysis and discussion to show that it can generalise well to other tasks and the criteria applied is generally a suitable one.

- The time warping error seems a suitable measure of the quality of the modelling but could the authors provide some detail on whether there is a significant difference in the performance for low errors? For example the authors state that the ESN with lowest error has a value of 11.37m while the ESN with best STM and PC performance has an error of 23.58m. Whilst the error is nearly twice the lowest, does it make a meaningful difference?

- It is typical to 'warm up' the ESN dynamics before inputting task information and this might be longer for the larger networks. Can the authors clarify whether they do this for the cart-pole in addition for the already discussed STM and PC tasks?

- With regards to literature, the authors should consider this article, Manneschi et al. "Exploiting multiple timescales in hierarchical echo state networks." *Frontiers in Applied Mathematics and Statistics* 6 (2021): 616658, also considers the timescales of an ESN in relevance to the timescales of the task albeit by using the leakage rate and spectral radius rather than the frequency response. It would be interesting to see how this relates to the successful and failure reservoirs in the submitted work.

- A more general comment, the main text is full of lists of values that make it hard to follow the flow of the discussion. These results would be better formatted as a table or similar and this would make the main text a little easier for the reader.

Version 1:

Reviewer comments:

Reviewer #1

(Remarks to the Author)

This revision improved the manuscript, but raises questions about the additions (pages 7-9). Why did the authors use suboptimal parameters? While the CR for ESN20s was 0.55, the graph for tau in Fig. 4A shows that there is a parameter region with a CR close to 1. Therefore, the results in this paper were for moderate CR and did not support the main hypothesis of this paper that reservoirs with higher CR perform better. The simulations need to be redone with optimal parameters. In addition, the advantages of the proposed method over parameter optimization using the gradient descent were not shown. This was because the authors manually adjusted the parameters. In the rebuttal letter, the authors discussed optimization using meta-heuristic algorithms, which was not done in this paper. Parameter optimization by some optimization method is mandatory for publication in the journal. For these reasons, we request that the paper should be re-submitted after re-experimentation.

The values of the maximum Lyapunov exponent were something wrong. If the maximum Lyapunov exponent is negative, the system is stable; if positive, it is chaotic. All values in Fig. 4C were positive, which means that the system was unstable.

Reviewer #2

(Remarks to the Author)

The authors have addressed almost all the points raised. I have concerns about the calculation of maximum Lyapunov exponent (MLE). By checking Fig. 4C, it seems that in almost all the parameter range, the value is positive, which means that the system is chaotic. Is the MLE correctly measured? If you are using the input-driven system, it should be called "conditional" MLE and it should basically behave similarly to that of ESP index. If there exists some uncertainties in measuring conditional MLE, it would be better to discard this analysis to avoid overall inconsistency.

Reviewer #3

(Remarks to the Author)

I thank the authors for the positive responses to my earlier review. I am glad to see that they have included more detail to fully explain and answer the queries that I had. In general the additions have strengthened the study and provided some clarity to the applicability of their approach. I still have some reservations about the generalisation of some of their conclusion to more complex tasks, I am happy that this represents a strong foundation that could be translated to other investigations.

Version 2:

Reviewer comments:

Reviewer #1

(Remarks to the Author)

I accept this paper on the condition that the authors add a limitation to the Discussion.

Reviewer #2

(Remarks to the Author)

The authors have addressed the raised points. In the revised manuscript, the authors mentioned, "... However, since no established method exists for calculating the CLE of an ESN, we adopted the ESPI as an alternative." This statement is misleading, since there exists several conventional methods for this (such as using Jacobian). I recommend the authors simply state as "For simplicity and clarity, we adopted the measure called ESPI..." or something like that.

We greatly appreciate the comments provided by the reviewer to improve the quality of this manuscript. We have done our best to address all the suggestions made by the reviewers and improved the manuscript accordingly. These revisions include the addition of Fig. 4, Tables 1 and 2, as well as a supplementary video. The following parts contain the original reviewers' comments and our actions taken for each comment. Responses having changes in the manuscript start with references [Page; Line] indicating the paragraph and section in the revised manuscript. All changes inside the manuscript regarding comments and questions are highlighted in Blue color.

Reviewers' comments:

Reviewer #1 (Remarks to the Author):

This study proposed a robot controller design using reservoir computing where reservoir parameters are screened so that the frequency of the robot (task) movements matches that of the reservoir dynamics, which is termed as robot-reservoir timescale alignment. From the data obtained when the robot was controlled by LQR, the reservoir learned the inverse dynamics of the robot, resulting in a controller that is more robust than LQR. Although a method for constructing a controller using a reservoir has already been proposed [16, Zhai et al., 2023], the idea of aligning the timescale of the controller (reservoir) and the controlled object (robot) is interesting. However, the timescale of the reservoir is mainly determined by the leakage parameter or time constant of unit dynamics, and the relationship between it and the alignment is unclear. A comparison with a method that optimize it directly is also needed. The following are major comments.

(1) The definition of inverse dynamics is unclear. Please clarify the learning task of the reservoir so that it can be understood by non-specialists in robotics.

- reply: [Page 17, Line 474~486 & Page 21, Line 593~597] Many thanks for your suggestions. In the new Sec.4.1, [Page 17, Line 474~486], we incorporated the robot's inverse dynamics using the Euler-Lagrange formulation, along with the desired system dynamics of various example controllers. The details of the training input and target is expressed in [Page 21, Line 593~597] to provide a clear explanation of the learning task.

(2) In this study, timescale alignment was proposed to simplify the adjustment of reservoir parameters. However, a method of optimizing the reservoir parameter

(leakage parameter and spectral radius) using a stochastic gradient descent method has also been proposed [A]. Identify the advantages of the proposed method compared to this existing method.

- reply: [Page 7~9, Line 199~234] Thank you for your comments. The objective of our study is to identify a metric capable of determining whether a suboptimal reservoir can be utilized for robotic control. In our evaluation, we aimed to empirically demonstrate the metric's effectiveness in distinguishing reliable reservoir controllers. This metric could potentially serve as an objective function in optimization methods, such as stochastic gradient descent, in future work. However, as illustrated in the referenced study [A, Fig. 2], the presence of multiple local minima within the search space suggests that the choice of the initial parameter set has a significant impact on the optimization process and its outcomes. The stochastic gradient descent method operates under two primary assumptions: (1) the cost function, in our case the control error, is convex with respect to the parameters, and (2) the cost function is deterministic. However, due to the nonlinear influence of hyperparameters on reservoir dynamics, and the dependence of the reservoir's control performance on the robot's nonlinear dynamics, there lacks mathematical proof confirming the convexity of the control error. Moreover, as shown in Fig. 6A, our results demonstrate that even with identical hyperparameter sets, randomly generated reservoirs exhibit diverse dynamics, leading to variations in control error. To overcome these challenges, alternative optimization approaches, such as meta-heuristic algorithms, may be employed to search for a global minimum. Since the proposed containment ratio metric does not require real-time deployment on a physical robot and can be tested offline, this approach offers a more time-efficient and cost-effective solution for searching.

- We introduced a new Fig. 4 to present a grid test analyzing how the containment ratio, an index from the timescale alignment test, is influenced by hyperparameters. The results indicate that certain hyperparameter sets can improve the containment ratio, suggesting that parameter optimization remains valuable for enhancing timescale alignment. For example, literature [Manneschi et al. "Exploiting multiple timescales in hierarchical echo state networks." *Frontiers in Applied Mathematics and Statistics* 6 (2021): 616658] employed gradient descent methods to optimize task performance based on reservoir timescale. However, as previously discussed, concerns remain regarding the fundamental assumptions of gradient descent methods when applied to ESNs. Therefore, we propose using the containment ratio to evaluate the timescale of individual ESNs, acknowledging that they may exhibit different dynamics even under identical hyperparameters. Additional discussions are added in [Page 7~9, Line 199~234], and how we select parameters are explained in [Page 7, Line 180~198].

(3) As mentioned above, the timescale of the reservoir strongly depends on the leakage parameter. The relationships between reservoir parameters and the containment ratio (CR) should be analyzed.

- reply: [Page 7, Line 199~211 & Fig.4] Thanks for your suggestions. In the new Fig. 4, We examined the relationship between hyperparameters and the containment ratio. The results indicate that the leakage parameter, which controls the reservoir's response speed to inputs, has a significant effect on the containment ratio, supporting our hypothesis that timescale plays a crucial role, as directly reflected in the results. Other factors, such as the number of nodes, spectral radius, input scaling, and bias scaling, also influence the containment ratio, while the link probability has no significant impact. Detailed discussions can be found in [Page 7, Line 199~211 & Fig.4]. Additionally, in subsequent sections, we added explorations about the relationship between the containment ratio, the echo state property index, and the maximum Lyapunov exponent. Please feel free to refer to those discussions for further insights.

(4) The spectral radius is the most important parameter for the performance of the reservoir. For general echo state networks, the best parameter value is about 1.0. However, as far as Fig. 5DE is concerned, its parameter search tick range is 0.3, which is too coarse. The spectral radius dependence of its performance is particularly strong as the network size increases. A more detailed parameter search, including around 1.0, is needed.

- reply: [Page 12, Line 301~304 & Fig.6] Thank you for your suggestions. We conducted a more detailed investigation of the spectral radius around 1.0 in Fig. 6, using a parameter search step size of 0.05. The analysis reveals that the dependency on the spectral radius increases as the number of nodes grows. The results indicate that a spectral radius of 1.4 generally yields the best performance for short-term memory and parity check tasks. However, increasing the spectral radius beyond 1 significantly destabilizes the reservoir's behavior. As shown in Fig. 4B, the echo state property index rises with increasing spectral radius, indicating that the reservoir's behavior becomes more sensitive to initial conditions. Based on these findings, we chose a spectral radius of 1.1 for our parameter set to balance stability and performance.

[A] Jaeger et al., "Optimization and applications of echo state networks with leaky-integrator neurons," *Neural Networks*, 20(3), 335-352, 2007.

Reviewer #2 (Remarks to the Author):

This paper investigates a reservoir computing approach for robot control, proposing a novel scheme called a timescale alignment. This scheme basically compares the FFT spectrum of the timeseries obtained by the target robotic systems and that obtained by the reservoir states, which takes the robotic timeseries as input, using the heuristic measure called containment ratio (CR). As a result, the scheme suggests that reservoirs that align their timescale with the robot show good control performance. Further systematic analyses of timescale aligned ESN are conducted, including measuring its STM and PC capacities. Overall, the topic is timely and the study includes interesting contribution to the field. Thus, I recommend the paper be accepted after addressing the following points (including major and minor points).

1. It is unclear how the hyperparameters of ESN are updated through the timescale alignment procedures. It is good to use CR to compare, but are these procedures performed through grid search of hyperparameters or are there any update rule proposed, such as gradient descent type scheme? I couldn't find clear descriptions about this point (or, if it was described somewhere, please stress it).

- reply: [Page 6, Line 159~164] Many thanks for your questions. We randomly generated 400 reservoirs with varying hyperparameters to identify a metric capable of determining whether a suboptimal reservoir can be used for robotic control, see [Page 5~7, Line 153~179]. In our evaluation, we empirically demonstrate that the proposed timescale alignment metric effectively distinguishes reliable reservoir controllers. This metric holds potential for future use as an objective function in optimization methods such as meta-heuristic algorithms. As suggested, we have highlighted the test set of reservoirs in the manuscript [Page 6, Line 159~164].

- For further analysis, we manually selected hyperparameters to align with the robot's timescale, whose mean CR is 0.55, larger than suggested threshold 0.4. In this process, hyperparameters were fixed. Detailed information on the parameter selection process is provided in an itemized list on [Page 7, Line 180~198].

- Additionally, the influence of hyperparameters on the CR is worth further exploration. To investigate this, we conducted a grid search, detailed in [Page 7~8, Line 199~211 & Fig.4A], to identify whether there are parameter sets that outperform those we manually selected. The results show that the number of nodes, leakage parameter, spectral radius, input scaling, and bias scaling all significantly affect CR, whereas the leakage parameter is the most sensitive. In contrast, link probability was found to have no significant impact on CR.

2. One important property in reservoir computing is the synchronization of input stream with reservoir dynamics, which is termed generalized synchronization [2a] or echo state property (ESP) [2b]. This property guarantees the reproducible response of the reservoir against identical input stream and is essential for reliable computation. It would be beneficial for readers if the authors investigate the relationship between timescale alignments and ESP. ESP can be characterized by ESP index using random input usually [2c] (In the current paper, we can use the input stream, coming from the robotic system, instead). For example, there should be some parameter regions that has ESP but less timescale alignment (opposite can also occur). This analysis will provide fundamental and interesting aspects to the characteristic of reservoir dynamics. Furthermore, it would be better to analyze conditional largest Lyapunov exponent of ESNs, if possible, which provides more accurate condition of the reservoir dynamics.

[2a] Lu, Z., Hunt, B. R., & Ott, E. (2018). Attractor reconstruction by machine learning. *Chaos: An Interdisciplinary Journal of Nonlinear Science*, 28(6).

[2b] Yildiz, I. B., Jaeger, H., & Kiebel, S. J. (2012). Re-visiting the echo state property. *Neural networks*, 35, 1-9.

[2c] Gallicchio, C. (2018). Chasing the echo state property. arXiv preprint arXiv:1811.10892.

- reply: [Page 7~8, Line 199~229 & Fig.4] Many thanks for your suggestions! We carefully reviewed the suggested literature [2a, 2b, 2c, and one additional paper 2d that provides a detailed explanation of the maximum Lyapunov exponent (MLE) for ESNs] and computed the echo state property index (ESPI) and MLE for our reservoirs. The results reveal that ESPI and the CR exhibit similar trends, indicating favorable reservoirs when varying the number of nodes, leakage parameter, spectral radius, input scaling, and link probability. However, ESPI and CR diverge when bias scaling is varied. Additionally, CR presents clearer and smoother trends compared to ESPI. While MLE shows a similar trend to CR and ESPI when varying the leakage parameter, it contradicts both when other hyperparameters are adjusted.

- Achieving a high CR requires the reservoir to exhibit stable behavior, which is associated with a low ESPI. When varying the bias, CR benefits from less node bias, as it becomes more input-driven, whereas ESPI remains unaffected by node bias since such bias does not influence the reservoir's stability. On the other hand, MLE measures the maximum separation rate of node behaviors, which is not directly related to timescale or behavioral stability. Moreover, the linear regression in the reservoir's output layer can either compress or expand the separation of signals, resulting in variations in the output MLE that are not reflective of the reservoir's internal MLE. Therefore, the reservoir's MLE may not directly impact control performance, which could explain its contradiction with CR and ESPI. Details about how to compute ESPI and MLE are provided in [Page 20, Line 566~586].

[2d] Pathak, Jaideep, et al. "Using machine learning to replicate chaotic attractors and calculate Lyapunov exponents from data." *Chaos: An Interdisciplinary Journal of Nonlinear Science* 27.12 (2017).

3. It would be helpful for readers if the authors attached the supplementary video on how the reservoir works on robot control.

- reply: Thank you for your suggestions. We have uploaded a supplementary video in the paper submission system. In addition, we also attached this video on GitHub, <https://github.com/Kyushudy/reservoircontroller.git>

4. The term "edge of chaos" is not used so much in recent days in reservoir computing community because it is misleading in multiple aspects. Please check Ref. [3a] for details.

[3a] H. Jaeger (2021): Foreword to the book *Reservoir Computing: Theory, Physical Implementations, and Applications* (K. Nakajima and I. Fischer, eds.), Springer Nature Singapore

- reply: Thank you for your comments. We reviewed [3a] and removed all references to the "edge of chaos" concept from this paper. Instead, we present the results data to demonstrate that ESNs generally perform better when the spectral radius is around 1, as shown in [Fig. 4 and Fig. 6].

5. Figure references are sometimes incorrect. For example, in Section 4.2, first sentence, Fig. 3C should be Fig. 2A? (or, at least, it should not be Fig. 3C). In Section 4.3, sixth sentence, is Fig. 3E correct? Please again check the figure reference in detail.

- reply: [Page 17, Line 488~489] Thank you for helping check the figure references. The reference to Fig. 3C has been corrected to Fig. 3D, which provides a detailed view of the node interconnections. Fig. 3E remains correct, as it displays the reservoir behavior prior to the FFT analysis. We have also reviewed all other figure, table, and equation references to ensure consistency throughout the paper.

Reviewer #3 (Remarks to the Author):

In this submission the authors present an approach for utilising echo-state network (ESN) reservoirs for model-free control of a robotic system. By implementing a

validation algorithm that tests the randomly created reservoirs for suitable matching of reservoir to the timescale of the robot motion the authors are able to focus on better performing reservoirs. The authors show that the ESN reservoir is able to control the agent within the cart-pole task, where the cart is required to follow a desired trajectory while maintaining the pole vertical.

The major claims of the paper are that timescale alignment of the reservoir and model dynamics can help filter out unsuitable reservoirs and that ESN reservoirs can be used as a robotic controller comparable to conventional models such as linear quadratic regulators (LQR). This is only demonstrated on the cart-pole task and performance appears to degrade with increasing reservoir size. Overall, whilst only a single demonstration of the reservoir controller, this work does present a novel filtering strategy and evaluation of the reservoir performance. Some aspects of the work needs further clarification as follows:

(1) The demonstration of the methodology is only on a task with a single joint and it is unclear how it would translate to more complicated tasks, e.g if the cart pole had multiple joints. Have the authors considered other robotic tasks and could they discuss this translation to more complication systems?

- reply: [Page 15~16, Line 432~447 & Page 8, Line 230~231] Thank you for your comments. In our manuscript we selected the cart-pole system, a classical underactuated system, to demonstrate how the timescale alignment test functions and how it informs controller design. Although the training process and control algorithms are relatively generalizable, the data collection strategy for capturing a robot's dynamics varies across different systems. Consequently, we did not test the proposed framework on multiple robots or systems, as doing so might shift the focus of the paper toward the design of data collection strategies or the evaluation of the reservoir's control capabilities. This challenge is common in the development of reservoir controllers, making it standard practice to validate proposed algorithms on a single robotic platform. For instance, in [Zhai, Z. M., Moradi, M., Kong, L. W., Glaz, B., Haile, M., & Lai, Y. C. (2023). Model-free tracking control of complex dynamical trajectories with machine learning. *Nature communications*, 14(1), 5698.], for the control of fully actuated robot using partially observed states, the authors tested controller only on a single robot.

- Although we tested our method on only one robot, we have developed approaches, such as timescale alignment, task learning, and the selection of reservoir hyperparameters, to be applicable across different robotic systems. The details of these methods are as follows:

- a) The timescale alignment process, as indicated by Eq. 14, estimates the robot's timescale based on the maximum area occupied by its state signals in the FFT graph

in Fig. 3C. Regardless of the number of degrees of freedom (DOF) a robot possesses, the same procedure as the cart-pole is applied to get its maximum area, which is used to calculate the containment ratio.

- b) The learning task involves an actuation test conducted prior to reservoir training to collect data on the robot's random movements. This data is then used to normalize the robot's state signals, which serve as inputs to the reservoir, see Sec. 4.4. Consequently, irrespective of the robot's actual state space, the reservoir operates on a normalized version of the state space, ensuring consistency in its manipulation on different robots.

- c) The reservoir hyperparameters can be randomly generated to find reservoirs with a high containment ratio for different robots. The only difference is the minimum number of reservoir nodes. As the input to the reservoir comprises both the robot's current and target states, the reservoir's number of nodes should be at least twice the robot's DOF.

- We added a discussion about how we design our approach to be applicable for different robots in [Page 15~16, Line 432~447], and an clarification to further emphasize the main objectives of the paper in [Page 8, Line 230~231].

(2) Could the authors clarify the computational complexity of the task? The small 10 node ESNs work well for the task, better than the larger networks, and usually with lower STM or PC metrics and it would be useful to have a deeper discussion on this. If the network simply needs to recall a step or two of input with minimal non-linearity to provide the correct signal to the cart-pole then it means it is harder to generalise the conclusions to broader control tasks. The concept that larger reservoirs inhibit the model-free control due to a larger transformation to invert relates to this (for example as discussed in ref 39) but further discussion and analysis of this would be beneficial.

- reply: [Page 15, Line 410~418 & Page 17, Line 474~486] Thank you for your questions. We have included the expression for the cart-pole inverse dynamics and the target system dynamics for control in [Page 17, Line 474~486]. The inverse dynamics equation is a second-order non-linear ODE with two variables, as shown by Eq. 5. The literature [Hollerbach, J. M. Robotics, Computer Simulations for. in Encyclopedia of Physical Science and Technology 275–281 (Elsevier, 2003). doi:10.1016/B0-12-227410-5/00667-0.] suggests that the computational complexity of this equation is $O(n^4)$, where n is the robot's degree of freedom. However, this measure of complexity does not directly correlate with the computational capacity of the reservoir, as the reservoir does not perform conventional floating-point operations. To more accurately assess computational complexity, we instead evaluate the memory and nonlinearity demands of the control task. According to Eq. 5, the inverse dynamics of the robot are governed by a second-order nonlinear ODE,

necessitating a high degree of nonlinearity in the reservoir. Furthermore, Eq. 6 shows that the control target involves a time delay of 1 second. Given that the actuation signals are updated every 0.01 seconds, the reservoir must effectively recall 100 timesteps to maintain reliable control.

- Performing a single-step short-term memory (STM) task does not imply that memory is limited to one step in control tasks, as the definition of STM and PC tasks are based on discrete time, binary input, while the control task is defined in continuous time, analog input. Based on the STM metric definition, values greater than 1 suggest that the reservoir memory extends beyond 10 seconds. Since all reservoirs we created exhibit an STM metric higher than 1, they possess sufficient memory for the robot control task. This explains why larger reservoirs, despite having better memory capacity, do not lead to improved control performance.

- It is noteworthy that the reservoir's memory exceeds the necessary requirements for control, indicating that the memory capacity is overqualified for the task. Furthermore, in addressing whether memory and nonlinearity are essential for control, we demonstrate in Fig. 5E that ESNs outperform LQRs, which are linear and lack memory. Relevant discussions on this topic have been added in [Page 15, Line 410~418].

(3) The timescale alignment evaluates the containment ratio of the reservoirs timescales to span that of the robots. This doesn't appear to consider the placement of the peaks in the reservoir frequency response. If the reservoirs fundamental frequency is close to be off resonance of the robot is there poor matching of the dynamics? Would it be better to test reservoirs to have a broader frequency response? While the method should be of interest to other researchers it needs some deeper analysis and discussion to show that it can generalise well to other tasks and the criteria applied is generally a suitable one.

- reply: [Page 15~16, Line 432~447]. Thank you for your questions. The containment ratio compares the FFT results of robot and reservoir behaviors after normalizing their amplitudes. In cases where only a few nodes are off resonance, such as the reservoir shown in Fig. 3F, where one node has a fundamental frequency near 0 Hz with extremely high amplitude, good timescale alignment and high control performance can still be maintained. This is because the linear regression in the reservoir's readout layer can filter out irrelevant nodes and generate stable outputs for robot control. However, if all nodes are off resonance, it results in poor timescale alignment and leads to low control performance. Among the 400 randomly generated ESNs we tested, we did not encounter any cases where the reservoir was off resonance yet had a high containment ratio or performed well in control. Based on this, we believe such cases are exceptionally rare.

- It would be beneficial to test reservoirs for a broader frequency response, and we have taken this into account. The cart-pole system, without external actuation, has its own fundamental frequency, which is a single value. To explore a broader frequency response, we provided inputs composed of a combination of three sinusoidal signals to the cart-pole system and evaluated its behavioral frequency range within 0~1 Hz, as shown in Fig.3BC. The reservoir was processed similarly, as shown in Fig.3EF. The timescale alignment test does not need to cover an overly broad frequency range, as its purpose is to efficiently distinguish hundreds of reservoirs in a short period. The results in Fig. 3H demonstrate that the current CR is effective in identifying reservoirs capable of achieving stable control.

- Following the timescale alignment test, we conducted a detailed frequency response analysis over a broader range of frequencies for our proposed reservoir controllers. In Fig. 5C, we evaluated the frequency response of the ESN controllers by testing their performance in controlling the cart-pole under target trajectories with frequencies ranging from 0.1 to 10 rad/s. The results indicate that the control error increases as the frequency rises. For frequencies below 1 rad/s, the control error remains small, demonstrating that ESN controllers respond well to low-frequency inputs. However, at 10 rad/s, the control error escalates to over 500m. For comparison, as shown in Fig. 5A, the sinewave tracking error of LQR2 is 159m, and its performance is already considered poor. This suggests that 10 rad/s serves as an appropriate upper limit for testing the frequency response of the controllers.

- We also performed a frequency response test for the behavior of individual nodes under sinusoidal input, as presented in Fig.7C. The frequency range spans from 0.1 rad/s to 100 rad/s, and we display both the magnitude and phase response of each node. The results indicate that each node exhibits a distinct frequency response, contributing to a broad overall frequency range that can effectively cover the robot's operational range. This variability in node responses enhances the reservoir's ability to manage diverse control tasks across different frequencies.

- We demonstrated the effectiveness of the timescale alignment test using a cart-pole system. However, applying this method to more complex robotic systems requires further study in the future research. We have included an extended discussion on how we design our approach to be applicable for different robots in [Page 15~16, Line 432~447].

(4) The time warping error seems a suitable measure of the quality of the modelling but could the authors provide some detail on whether there is a significant difference in the performance for low errors? For example the authors state that the ESN with lowest error has a value of 11.37m while the ESN with best STM and PC performance has an error of 23.58m. Whilst the error is nearly twice the lowest, does it make a meaningful difference?

- reply: [Page 9, Table 1] Thanks for your question. We have added a new Table 1 to display the control errors of both LQRs and ESNs. By examining Table 1 alongside Fig. 5AB, the exact robot trajectories under different errors can be visualized. For example, LQR3's control error for the Lorenz tracking task is 35.99m, which is represented in the phase diagram as a significantly larger figure-eight shape compared to the target. The Lorenz tracking error for ESN50 is 11.37m, while for ESN500 it is 23.58m. However, due to the complexity of the trajectory and the slow accumulation of error, this difference is not immediately apparent in the phase diagram. In contrast, the sinewave tracking task more clearly illustrates the difference between ESN50 and ESN500. As shown in Fig. 5A, ESN500 exhibits a large initial bias, while ESN50 does not, leading to an additional 5m of control error for ESN500.

(5) It is typical to 'warm up' the ESN dynamics before inputting task information and this might be longer for the larger networks. Can the authors clarify whether they do this for the cart-pole in addition for the already discussed STM and PC tasks?

- reply: [Page 10, Line 244~246 & Page14, Line 381~382] Many thanks for your suggestions. The robot's and reservoir's initial state is set to zero for all values by default, and both are activated at time 0 seconds without any warm-up period. We implemented this approach because, when testing the basin of attraction in Fig. 5D, the robot begins at varying unstable angles. If the reservoir cannot immediately stabilize the robot, the control task will fail. However, this setup also introduces an initial bias when activating reservoir control, as seen in Fig. 5A, where ESN500 shows a significant initial bias. How to properly warm up the reservoir for control tasks remains an open question. We have added clarifications regarding the initial conditions of the reservoirs and robots in [Page 10, Line 244~246], along with related discussions in [Page14, Line 381~382].

(6) With regards to literature, the authors should consider this article, Manneschi et al. "Exploiting multiple timescales in hierarchical echo state networks." *Frontiers in Applied Mathematics and Statistics* 6 (2021): 616658, also considers the timescales of an ESN in relevance to the timescales of the task albeit by using the leakage rate and spectral radius rather than the frequency response. It would be interesting to see how this relates to the successful and failure reservoirs in the submitted work.

- reply: [Page 7&8, Line 199~211 & Fig. 4] Thank you for your comments. We introduced a new Fig. 4 to present a grid test analyzing how the containment ratio is influenced by hyperparameters. The results in [Page 7&8, Line 199~211] indicate that reducing the leakage rate and spectral radius can improve the containment ratio, which lead to lower failure rates of reservoir controllers as shown in Fig. 3H. This

suggests that parameter optimization remains valuable for enhancing timescale alignment.

- The suggested literature employs a gradient descent method to tune the leakage rate and spectral radius for optimal task performance, where the timescale is approximated using the minimum, maximum, and most probable (peak of the distribution) timescales of the reservoir. According to Eq. 26 in the literature, the cost function is based on the error between the reservoir output and the target. However, there are fundamental concerns about gradient descent methods applying on reservoir controller parameter optimization. The stochastic gradient descent method operates under two primary assumptions: (1) the cost function, in our case the control error, is convex with respect to the parameters, and (2) the cost function is deterministic. However, due to the nonlinear influence of hyperparameters on reservoir dynamics, and the dependence of the reservoir's control performance on the robot's nonlinear dynamics, there lacks mathematical proof confirming the convexity of the control error. Moreover, as shown in Fig. 6A, our results demonstrate that even with identical hyperparameter sets, randomly generated reservoirs exhibit diverse dynamics, leading to variations in control error. To overcome these challenges, alternative optimization approaches, such as meta-heuristic algorithms, may be employed to search for a global minimum. Since the proposed containment ratio metric does not require real-time deployment on a physical robot and can be tested offline, this approach offers a more time-efficient and cost-effective solution for searching.

(7) A more general comment, the main text is full of lists of values that make it hard to follow the flow of the discussion. These results would be better formatted as a table or similar and this would make the main text a little easier for the reader.

- reply: [Page 9, Table 1 & Page 12, Table 2] Thank you very much for your suggestions. We have transferred most of the values from the main text into Table 1 and Table 2 to improve readability, and the main text has been simplified accordingly. This allows for a clearer presentation of data, making it easier for readers to access and interpret key results at a glance. The edited main texts are in [Line 253~255, 260~263, 273~276, 279~282, 286~294, 301~304, 328~338].

We greatly appreciate the second round comments provided by the reviewer to improve the quality of this manuscript. We have done our best to address all the suggestions made by the reviewers and improved the manuscript accordingly. These revisions include the addition of Fig. 5 and the removal of Fig. 4C. The following parts contain the original reviewers' comments and our actions taken for each comment. Responses having changes in the manuscript start with references [Page; Line] indicating the paragraph and section in the revised manuscript. All changes inside the manuscript regarding comments and questions are highlighted in Blue color.

Reviewers' comments:

Reviewer #1 (Remarks to the Author):

This revision improved the manuscript, but raises questions about the additions (pages 7-9). Why did the authors use suboptimal parameters? While the CR for ESN20s was 0.55, the graph for tau in Fig. 4A shows that there is a parameter region with a CR close to 1. Therefore, the results in this paper were for moderate CR and did not support the main hypothesis of this paper that reservoirs with higher CR perform better. The simulations need to be redone with optimal parameters. In addition, the advantages of the proposed method over parameter optimization using the gradient descent were not shown. This was because the authors manually adjusted the parameters. In the rebuttal letter, the authors discussed optimization using meta-heuristic algorithms, which was not done in this paper. Parameter optimization by some optimization method is mandatory for publication in the journal. For these reasons, we request that the paper should be re-submitted after re-experimentation.

- reply: [Page 8~9, Line 219~253 & Fig. 5] Thank you for your questions. To address the reviewer's comment, we conducted hyperparameter optimization as suggested to maximize CR and demonstrate how the control success rate improves with increasing CR, as illustrated in the new Fig. 5. Since reservoirs are randomly generated based on specified hyperparameters, the CR is inherently non-deterministic in the cart-pole control task. To address this, we employed Bayesian optimization, which is well-suited for non-deterministic cost functions. To reduce uncertainties, we performed optimization across 60 independent trials, with each trial consisting of 120 iterations to allow CR to converge. The results reveal that at the 1st iteration, the ESNs are nearly randomly generated, yielding a mean CR of 0.18 and

a control success rate of 10%. By the 60th iteration, the mean CR increased significantly to 0.91, accompanied by a corresponding rise in the control success rate to 70%. Both the CR and the success rate converged after the 60th iteration. These findings confirm that reservoirs with higher CR exhibit a higher success rate in the cart-pole tracking control task.

- Furthermore, we selected the optimal ESN identified by Bayesian optimization, which achieved the highest CR of 1.0. This ESN exhibited a Lorenz trajectory tracking error of 16.60 m. In comparison, the best manually designed ESN20 achieved a slightly lower error of 12.32 m. These results indicate that both Bayesian optimization and manual design methods are capable of identifying reservoir controllers with high CR, leading to reduced control errors. However, a notable discrepancy was observed in the spectral radius (SR) of the optimal ESN, which was 18.99—substantially exceeding the $SR < 1$ threshold typically recommended by the echo state property. This suggests a high likelihood of instability in the optimal ESN. To evaluate this further, we generated 100 ESNs using the optimal hyperparameters identified through Bayesian optimization and found that only 38% succeeded in the control task. In contrast, the manually designed ESN20 reservoirs achieved a 100% success rate, as shown in Table 2. This highlights a key limitation of optimization methods: because the relationship between hyperparameters and control performance is non-deterministic, the optimal result identified by optimization may represent a fortunate outlier that performs well under specific conditions but is not robust across multiple trials. Furthermore, optimization methods are susceptible to suboptimal solutions, as there is no guarantee of convexity in the cost function. For instance, Fig. 5 shows considerable CR variance at the 120th iteration, indicating that many independent trials become trapped in suboptimal solutions. Therefore, optimization methods may struggle to consistently identify hyperparameter sets that reliably generate high-performing reservoir controllers.

- In summary, due to inherent uncertainties in the cost function, optimization methods may fail to identify the global minimum or the most reliable hyperparameter sets that consistently succeed in control tasks. To address this limitation, we propose an alternative approach: designing hyperparameters guided by timescale alignment, with a recommended $CR > 0.4$. While these hyperparameters may appear suboptimal (e.g., the CR for ESN20s is 0.55), they adhere to additional constraints, such as the echo state property. This adherence enhances the robustness of the controller, ultimately resulting in a higher success rate.

- The new Fig. 5 shows that reservoirs with higher CR have higher success rate, and we put relative analysis in [Page 8~9, Line 219~253].

The values of the maximum Lyapunov exponent were something wrong. If the maximum Lyapunov exponent is negative, the system is stable; if positive, it is

chaotic. All values in Fig. 4C were positive, which means that the system was unstable.

- reply: [Page 21, Line 590~594] Thank you for your comments. Since our ESNs are input-driven systems, the use of the maximum Lyapunov exponent (MLE) is indeed inappropriate, as MLE is designed for autonomous systems and would always indicate instability for our ESN. The correct alternative is the conditional Lyapunov exponent (CLE), which directly aligns with the concept of the echo-state property index (ESPI), which was already illustrated in Fig. 4B. Therefore, we have decided to remove all references to MLE, including Fig. 4C and the corresponding paragraphs previously located in [Page 7~8, Line 211~218]. To provide clarity, we have added a brief explanation of the relationship between ESPI and CLE in [Page 21, Line 590~594]. This adjustment ensures consistency and better alignment with our input-driven ESNs.

Reviewer #2 (Remarks to the Author):

The authors have addressed almost all the points raised. I have concerns about the calculation of maximum Lyapunov exponent (MLE). By checking Fig. 4C, it seems that in almost all the parameter range, the value is positive, which means that the system is chaotic. Is the MLE correctly measured? If you are using the input-driven system, it should be called "conditional" MLE and it should basically behave similarly to that of ESP index. If there exists some uncertainties in measuring conditional MLE, it would be better to discard this analysis to avoid overall inconsistency.

- reply: [Page 21, Line 590~594] Thank you for your valuable comments. Following your suggestions, we have removed all contents about MLE in this manuscript as it is not applicable to our reservoir controller scenario. MLE is a tool designed to analyze the stability of autonomous systems that are not influenced by external inputs. While methods exist for applying MLE to analyze the dynamical properties of ESNs [2a, 2b], these cases involve ESNs emulating autonomous systems such as the Lorenz system. In our control task, however, the ESNs are input-driven, as they process robot states as inputs and produce corresponding control actions. As a result, MLE is not applicable in our scenario, and the appropriate alternative is the CLE [2c, 2d], which evaluates the stability of systems driven by external chaotic signals. This concept aligns directly with the ESPI, which measures the sensitivity of reservoir dynamics to initial conditions when subjected to identical inputs. Despite this conceptual similarity, there is currently no formal method to compute the CLE specifically for ESNs. Consequently, we have discarded the MLE analysis and

included a brief discussion in [Page 21, Line 590~594] to clarify the relationship between ESPI and CLE for readers.

[2a] Lu, Z., Hunt, B. R., & Ott, E. (2018). Attractor reconstruction by machine learning. *Chaos: An Interdisciplinary Journal of Nonlinear Science*, 28(6).

[2b] Pathak, J., Lu, Z., Hunt, B. R., Girvan, M., & Ott, E. (2017). Using machine learning to replicate chaotic attractors and calculate Lyapunov exponents from data. *Chaos: An Interdisciplinary Journal of Nonlinear Science*, 27(12).

[2c] Pecora, L. M., & Carroll, T. L. (1991). Driving systems with chaotic signals. *Physical review A*, 44(4), 2374.

[2d] Pecora, L. M., Carroll, T. L., Johnson, G. A., Mar, D. J., & Heagy, J. F. (1997). Fundamentals of synchronization in chaotic systems, concepts, and applications. *Chaos: An Interdisciplinary Journal of Nonlinear Science*, 7(4), 520-543.

Reviewer #3 (Remarks to the Author):

I thank the authors for the positive responses to my earlier review. I am glad to see that they have included more detail to fully explain and answer the queries that I had. In general the additions have strengthened the study and provided some clarity to the applicability of their approach. I still have some reservations about the generalisation of some of their conclusion to more complex tasks, I am happy that this represents a strong foundation that could be translated to other investigations.

- reply: Thank you for your feedback. Expanding reservoir controllers to more complex robotic systems is a fascinating and promising research direction, and we are excited to explore this avenue in future work. We sincerely appreciate your assistance in improving this manuscript and thank you once again for your support!

We greatly appreciate the second round comments provided by the reviewer to improve the quality of this manuscript. We have done our best to address all the suggestions made by the reviewers and improved the manuscript accordingly. These revisions include the addition of Fig. 5BC. The following parts contain the original reviewers' comments and our actions taken for each comment. Responses having changes in the manuscript start with references [Page; Line] indicating the paragraph and section in the revised manuscript. All changes inside the manuscript regarding comments and questions are highlighted in Blue color.

Reviewers' comments:

Reviewer #1 (Remarks to the Author):

The authors have appropriately added the experiment; better success rates through optimization for CR make this paper worthwhile. However, the optimization has led to inappropriate reservoir conditions (overfitting?). The fact that the performance was worse than that of the ESN20s with CR of 0.55 limits the claims of this paper. That is, increasing CR does not necessarily improve the performance of the robot control. I suggest that another criteria or constraint on the appropriate reservoir activity, e.g., ESPI, is included in the objective function of the optimization. In light of the above, the discussion section should be revised to reflect this result. In particular, the first paragraph should include this limitation. After this revision, I would like to accept this paper for publication.

- reply: [Page 4, Line 106~110] [Page 9, Line 251~268 & Fig. 5] [Page15, Line 385~393, Line 408~424] Thank you for your suggestions. To address the reviewer's comment, we conducted additional hyperparameter optimization for ESPI (Fig. 5B) and ESPI-CR (Fig. 5C), and added the limitations of CR optimization in the first and third paragraphs of the discussion section.

The results show that solely optimizing CR (Fig. 5A) does not consider the echo-state property, therefore only 38% of new ESNs generated by the optimum hyperparameter set achieved successful control. On the other hand, Fig. 5B demonstrates that optimizing solely for ESPI results in a low control success rate of 30%, as ESPI does not account for the robot-reservoir timescale alignment. To address these limitations, we conducted a combined approach that considers both CR and ESPI during hyperparameter design. Fig. 5C shows that by increasing CR and decreasing ESPI through Bayesian optimization, the control success rate increases to 75%, and 92% of the new ESNs generated by the optimum

hyperparameter set achieved robot control. Meanwhile, Sec.2.3 shows that hyperparameter sets manually designed using the same principles achieve a 100% control success rate.

- The new Fig. 5 shows the optimization result of CR, ESPI, and ESPI-CR. Relative results are illustrated in [Page 9, Line 251~268]. The limitations about solely optimizing CR are discussed in [Page15, Line 385~393, Line 408~424] and [Page 4, Line 106~110].

Reviewer #2 (Remarks to the Author):

The authors have addressed the raised points. In the revised manuscript, the authors mentioned, "... However, since no established method exists for calculating the CLE of an ESN, we adopted the ESPI as an alternative." This statement is misleading, since there exists several conventional methods for this (such as using Jacobian). I recommend the authors simply state as "For simplicity and clarity, we adopted the measure called ESPI..." or something like that.

- reply: [Page 22, Line 630~631] Thank you for your comments. We edited the relative sentences as suggested in [Page 22, Line 630~631].